# SEMANTIC-ANCHORED, CLASS VARIANCE-OPTIMIZED CLUSTERING FOR ROBUST SEMI-SUPERVISED FEW-SHOT LEARNING

## ABSTRACT

Few-shot learning has been extensively explored to address problems where the amount of labeled samples is very limited for some classes. In the semi-supervised few-shot learning setting, substantial quantities of unlabeled samples are available. Such unlabeled samples are generally cheaper to obtain and can be used to improve the few-shot learning performance of the model. Some of the recent methods for this setting rely on clustering to generate pseudo-labels for the unlabeled samples. Since the effectiveness of clustering heavily influences the labeling of the unlabeled samples, it can significantly affect the few-shot learning performance. In this paper, we focus on improving the representation learned by the model in order to improve the clustering and, consequently, the model performance. We propose an approach for semi-supervised few-shot learning that performs a class-variance optimized clustering coupled with a cluster separation tuner in order to improve the effectiveness of clustering the labeled and unlabeled samples in this setting. It also optimizes the clustering-based pseudo-labeling process using a restricted pseudo-labeling approach and performs semantic information injection in order to improve the semi-supervised few-shot learning performance of the model. We experimentally demonstrate that our proposed approach significantly outperforms recent state-of-the-art methods on the benchmark datasets. To further establish its robustness, we conduct extensive experiments under challenging conditions, showing that the model generalizes well to domain shifts and achieves new state-of-the-art performance in open-set settings with distractor classes, highlighting its effectiveness for real-world applications. Here is the repo link: Anonymous Repository Link

## 1 INTRODUCTION

Modern deep networks often surpass human performance but demand ever deeper architectures and vast labeled datasets, whose annotation is costly and time-consuming. This data bottleneck motivates few-shot learning, where models must generalize from only a handful of examples. Therefore, researchers have been looking at few-shot learning (FSL) as a solution. Few-shot learning tackles new classes with only a handful of labeled examples by first training a feature extractor on richly annotated base classes and then transferring those learned representations to extract meaningful embeddings for the scarce few-shot samples. The performance of existing FSL methods is still limited due to the extremely limited availability of labeled samples(Dong et al., 2024; Sun et al., 2019). Unlabeled data is often plentiful and easy to collect, so incorporating large pools of unlabeled examples alongside the scarce few-shot samples, known as semi-supervised few-shot learning, can substantially boost performance. In this paper, we tackle the semi-supervised few-shot learning (SSFL) problem.

Researchers have proposed multiple approaches to address the SSFSL problem, such as clustering to assign pseudo-labels to unlabeled samples and utilizing them to improve the FSL performance (Rodríguez et al., 2020; Ling et al., 2022). Clustering relies heavily on the quality of representation learned by the network, especially on the base classes, since this ensures that newly observed images, even from unseen classes, are far away from images of other classes and close to images from the same class in the representation space. A model with better quality representation will facilitate

better clustering of extracted features and consequently lead to better pseudo-label assignment to the unlabeled samples (refer to Table 5).

We propose a novel SSFSL approach in this work. Specifically, we use a class variance optimized clustering (CVOC) process to cluster given images and use a semantic injection network to refine the cluster centers during inference by incorporating semantic information derived from the class labels. The proposed CVOC process optimizes intra- and inter-class separation and utilizes a cluster separation tuner to achieve improved clustering (refer to Table 5). We use the refined cluster centers to perform restricted pseudo-labeling of the unlabeled samples and incorporate all labeled samples to make predictions for the query samples (see Sec. 4). We perform experiments on multiple benchmark datasets and demonstrate that our proposed approach significantly outperforms the compared approaches.

The key contributions of this paper are as follows:

1. We propose a novel semi-supervised few-shot learning approach that outperforms the recent methods in this setting, as shown in Tables **??**, **??**, 3.

2. We experimentally demonstrate how our approach improves the pseudo-labeling accuracy, as shown in Table 5, and is, therefore, able to outperform the compared methods.

3. We experimentally demonstrate how incorporating semantic features of classes and managing the intra-class compactness and inter-class separation leads to better feature representations in the embedding space, thereby improving the performance of the model, as shown in Table 4. We also provide a novel class description generation pipeline (see Section A.4.4), which overcomes raw class name noise, poor label description,etc.

4. Furthermore, we validate the robustness and generalization capabilities of our approach under challenging realistic scenarios, including new state-of-the-art performance in distractive semi-supervised few-shot learning settings and strong cross-domain few-shot learning results. Refer to Section A.1.

## 2 Related Works

### 2.1 Few-shot Image Classification

Few-shot learning (FSL) methodologies (Kozerawski & Turk, 2018; Yoo et al., 2018; Rusu et al., 2018; Gidaris & Komodakis, 2019; Lee et al., 2019) generally pre-train the model using extensive data from the base classes and subsequently modify the pre-trained model for recognizing novel categories. (Snell et al., 2017) utilizes an average feature prototype of the classes with very few samples and performs predictions by identifying the nearest prototype. TADAM (Oreshkin et al., 2018) adapts the features of the few-shot class images using the few support images available for such classes. (Ji et al., 2021) improves upon prototypical network using an importance-weighting strategy for the different support samples of a class and makes the prototypes more informative. Meta-learning methods learn to quickly adapt to new classes using very few samples, and these methods involve training a model at a specific level as well as at an abstract level, referred to as learning to learn. (Yu et al., 2020). Researchers have proposed various meta-learning-based FSL approaches, such as (Finn et al., 2017; Li et al., 2017). .

### 2.2 Pseudo-labeling based Semi-supervised Few-shot Methods

The semi-supervised few-shot learning (FSL) setting is a modified FSL setting with access to unlabeled samples. Several semi-supervised FSL methods assign pseudo-labels to unlabeled samples and then consider these samples as labeled samples for the few-shot classification task (Lazarou et al., 2021; Wu et al., 2018; Yu et al., 2020). However, if some of the pseudo labels are incorrect, the performance of the model is degraded. Some methods utilize label propagation to generate pseudo-labels (Liu et al., 2018). (Li et al., 2019) selectively sample high-confidence labels, but treat each sample independently and miss out on broader relational context. (Huang et al., 2021b) attempts to unify pseudo-labeled data in a single metric space, but also does not consider the inter-sample relationships. Poisson Transfer Network (PTN) (Huang et al., 2021a) incorporates some relational understanding between labeled and unlabeled samples. However, it lacks the capacity to

effectively mine inter- and intra-class relationships. Instance Credibility Inference (ICI) (Wang et al., 2020) ranks pseudo-labels based on credibility and sparsity, but neglects the issue of low-quality pseudo-labels. Cluster-FSL (Ling et al., 2022) employs a multi-factor clustering (MFC) strategy, which synthesizes various information sources from labeled data to produce both soft and hard pseudo-labels. It pseudo-labels selected unlabeled samples. However, we experimentally observed that the pseudo-labeling strategy employed by it led to most of the unlabeled images becoming eligible for pseudo-labeling, which would degrade the clustering process with outlier samples or incorrectly pseudo-labeled samples. Cluster-FSL's limitations are rooted in its inadequate handling of inter-class and intra-class relationships, which restricts its ability to differentiate effectively between distinct classes and to capture subtle variations within individual classes. This leads to unlabeled samples with incorrect pseudo-labels being considered as labeled samples and consequently limiting the model performance.

In this work, we address these limitations through a dual-novelty framework. First, we introduce a class-variance optimized clustering (CVOC) approach, together with a cluster separation tuner (CST), which acts as a geometric regularizer that aligns the embedding space according to both inter- and intra-class relationships via targeted loss functions. This design yields a cleaner manifold that reduces overlap between classes and pulls in-class samples closer, which is critical for robustness under open-set distractors and domain shift. Second, we propose a Semantic Injection Module that learns a shared non-linear latent manifold between visual and textual features, going beyond standard linear projectors that suffer from hubness and misalignment. This module is model-agnostic and pre-trained separately on text and image features, so it can be plugged into any SS-FSL backbone without interfering with few-shot training, enabling test-time semantic alignment in a zero-leakage fashion. Our approach also incorporates high-quality semantic class descriptions and restricted pseudo-labeling, allocating pseudo-labels exclusively to unlabeled samples with low-entropy predictions, thereby reducing noisy labels while keeping inference latency low. Taken together, these components not only achieve state-of-the-art pseudo-label accuracy among clustering-based SS-FSL methods, but also position our framework as a forward-looking solution for realistic scenarios dominated by distractors and domain shifts, rather than idealized clean benchmarks.

## 3 PROBLEM SETTING

In the few-shot learning (FSL) setting, there are two sets of non-overlapping classes: the base classes $C_b$ and the novel/few-shot classes $C_f$. The base classes have a sufficient number of training samples, while the novel classes have very few training samples. The dataset is split into the train set $D_{tr}$, the validation set $D_{va}$, and the test set $D_{tt}$. The train set $D_{tr}$ only contains samples from the base classes, while the validation $D_{va}$ and test $D_{tt}$ sets contain samples from the novel classes without any overlap between the classes in $D_{va}$ and $D_{tt}$. Data is provided as $N$-way $K$-shot episodes/tasks, where $N$ refers to the number of classes in the episode, and $K$ refers to the number of support/training samples from each of these classes. The support set can be denoted as $S = (x_i^s, y_i^s)_{i=0}^{N \times K}$. The episode also contains $T$ randomly selected query/test samples from each of these classes on which the model can be evaluated. The query set can be denoted as $Q = (x_i^q, y_i^q)_{i=0}^{N \times T}$. The episodes from the train set contain a support set $S$ and a query set $Q$ from $N$ randomly selected classes from among the classes in the train set. Similarly, the episodes from the test set contain a support set $S$ and a query set $Q$ from $N$ randomly selected classes in the test set. In this paper, we address the semi-supervised FSL setting. In this setting, apart from the above episodic data, we also have access to $u$ randomly selected samples from the classes in the episode that are not labeled. The unlabeled set can be denoted as $U = (x_i^a)_{i=0}^{N \times u}$. This leads to a setting where we have very few training samples for the classes in the episode, but a good quantity of unlabeled samples.

## 4 METHODOLOGY

### 4.1 COMPONENTS

The proposed approach involves training a visual feature extractor network ($M$) and a semantic injection network ($S$). The visual feature extractor $M$ is a simple deep neural network that takes an image as input, and once trained, it should be able to extract good features from images. The semantic

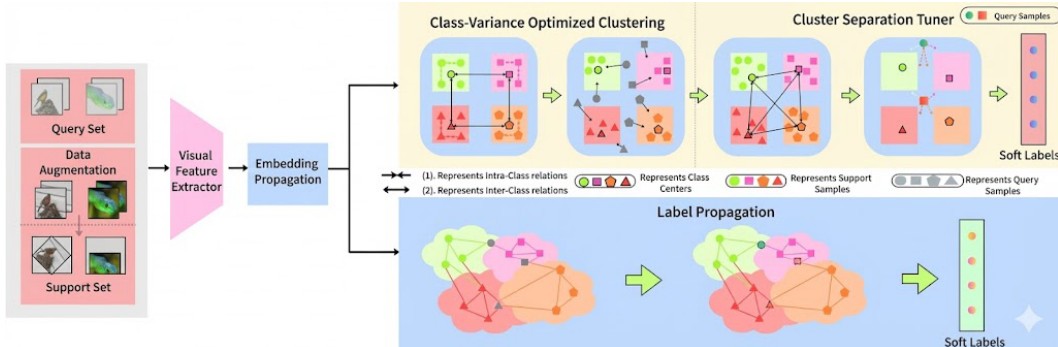

Figure 1: Illustration of the fine-tuning phase. The support and query sets are passed through a visual feature extractor and embedding propagation. The resulting embeddings are processed in parallel by two modules: (i) Class-Variance Optimized Clustering (CVOC), which refines cluster assignments by leveraging intra- and inter-class distance regularization, followed by the Cluster Separation Tuner (CST) to enhance prototype separation before generating soft labels for query samples; and (ii) Label Propagation (LP), which also propagates support labels to query embeddings.

injection network contains an encoder module ($E_s$) and a decoder module ($D_s$). The encoder module $E_s$ takes as input the prototype feature of a class concatenated with the semantic feature of the same class and maps it to a joint embedding space, while the decoder module $D_s$ reconstructs the input to the encoder module from the joint embedding. Once trained, the semantic injection network $S$ should be able to map the input features to a joint embedding space where the joint feature embedding is close to the image feature prototype of the same class.

## 4.2 PRE-TRAINING THE VISUAL FEATURE EXTRACTOR

The visual feature extractor $M$ is first pre-trained on the train set $D_{tr}$, containing the base classes, in a batch-wise supervised setting. Following various existing methods, we train $M$ simultaneously on the image label classification task and the rotation prediction task (see Eq. 1), since this joint training has been shown to improve the generalization power of the network. We add an image label classification head $F_c$ and a rotation angle classification head $F_r$ to $M$ for this training process. The images are rotated by four fixed angles and the model predicts the class label using $F_c$ and the rotation angle label using $F_r$. The pretraining loss $L_{pt}$ is defined below:

$$L_{pt}(x_i, y_i, r_i) = L_{\text{ce}}\big(F_c\big(M(x_i)\big), y_i\big) + L_{\text{ce}}\big(F_r\big(M(x_i)\big), r_i\big). \tag{1}$$

where, $y_i$ and $r_i$ refer to the image label and rotation label for image $x_i$, respectively. $L_{\text{ce}}$ refers to the cross-entropy loss.

## 4.3 FINETUNING THE VISUAL FEATURE EXTRACTOR WITH CLASS-VARIANCE OPTIMIZED CLUSTERING

After pre-training the visual feature extractor $M$, we fine-tune it on episodes from the train set $D_{tr}$. In the few-shot regime, with only a handful of labeled support examples in each episode, aggressive augmentations greatly increase input diversity and prevent the model from simply memorizing the small support set. Following recent SSFSL methods (Ling et al., 2022), we begin by applying RandAugment (Cubuk et al., 2020) to every support image to guard against over-fitting. After extracting embeddings for both these augmented support samples and the query set with our visual feature extractor $M$, we refine all features using embedding propagation (Rodríguez et al., 2020; Ling et al., 2022), a technique commonly used in SSFSL to obtain manifold-smoothed embeddings. This process is described in A.4.1 in the Appendix. Finally, the propagated representations are fed into our Class-Variance Optimized Clustering (CVOC) module (Fig. 1). Then, based on the support set, the CVOC module, and label propagation (LP), we obtain the soft labels of the query set samples. Label

propagation (Iscen et al., 2019) is a graph-based technique commonly used in SSFSL to "propagate" labels from labeled samples to unlabeled samples, utilizing the similarity between the samples as weights for edges between the graph nodes. It is also discussed in A.4.1 in the Appendix.

### 4.3.1 CLASS-VARIANCE OPTIMIZED CLUSTERING AND CLUSTER SEPARATION TUNER

The proposed class-variance optimized clustering procedure utilizes an improved clustering procedure and a cluster separation tuner (CST) to get a more robust clustering of the labeled and unlabeled image features in the episode. For each class label $c \in \{1, \ldots, C\}$, we construct a class-specific factor dictionary $\mathbf{F}_c$ consisting of all support embeddings for that class together with its initial prototype, defined as the mean embedding of the corresponding support samples:

$$\mathbf{F}_c = \left[ s_1^{(c)}, \ldots, s_K^{(c)}, \ P_0(c) \right] \in \mathbb{R}^{d \times (K+1)},$$

where $s_k^{(c)} \in \mathbb{R}^d$ denotes the $k$-th support embedding and $P_0(c)$ is the initial prototype. Given a feature vector $x$, we compute its coefficient vector by solving the regularized least-squares problem:

$$\beta_c^*(x) = \arg\min_{\beta} \ \|x - \mathbf{F}_c \beta\|_2^2 + \lambda \|\beta\|_2^2,$$

with $\lambda = 0.01$. The reconstruction distance is then

$$d_{\text{rec}}(x, c) = \|x - \mathbf{F}_c \beta_c^*(x)\|_2^2,$$

which measures how well $x$ can be represented using the support embeddings and prototype of class $c$.

Furthermore, the coefficient vector decomposes as

$$\beta_c^*(x) = \left[ \beta_{c,1}^*(x), \beta_{c,2}^*(x), \ldots, \beta_{c,N}^*(x) \right]^\top,$$

where each sub-vector $\beta_{c,j}^*(x) \in \mathbb{R}^{(K+1) \times 1}$ corresponds to the $j$-th cluster within the class, offering a richer representation of intra-class variability by allocating different coefficient subsets to different clusters.

The reconstruction-based distance captures the true shape and spread of each category by leveraging both the cluster center and representative labeled samples. This richer, multi-prototype view helps us account for within-category heterogeneity and yields more accurate, robust assignments for unlabeled data compared to traditional clustering techniques. However, $d_{\text{rec}}$ alone may be ambiguous when class subspaces overlap. Therefore, we regularize it with two Euclidean terms, i.e., $L_{\text{intra}}$ and $L_{\text{inter}}$. Let $P(c)$ denote the current prototype of class $c$ (updated after every assignment step) and $\text{ep}\big(M(x)\big)$ denote the embedding propagated feature of $x$ as in Rodríguez et al. (2020). Using $\mathcal{S}$ to denote the support set (expanded support set obtained through augmentation), we define

$$L_{\text{intra}} = \frac{1}{|\mathcal{S}|} \sum_{(x_i, y_i) \in \mathcal{S}} \left\| \text{ep}\big(M(x_i)\big) - P(y_i) \right\|_2^2, \tag{2}$$

$$L_{\text{inter}} = \frac{1}{|\mathcal{S}|} \sum_{(x_i, y_i) \in \mathcal{S}} \frac{1}{C-1} \sum_{\substack{j=1 \\ j \neq y_i}}^{C} \left\| \text{ep}\big(M(x_i)\big) - P(j) \right\|_2^2, \tag{3}$$

Here $C$ is the number of classes in the episode. Considering an image $x_i$ and label $y_i$, $P(y_i)$ denotes the prototype of the class $y_i$, and $P(j)$ denotes the prototype of some *other class* $j$ present in the episode. The *intra-class* term $L_{\text{intra}}$ computes, for every support sample, the squared Euclidean distance between its embedding (after embedding propagation) and the prototype of its own class, and then averages these distances over the entire support set. Conversely, the *inter-class* term $L_{\text{inter}}$ takes each support sample, measures its squared distance to *every other* class prototype, averages those distances across the $C-1$ "foreign" classes, and finally averages the resulting values over all support samples.

The combined distance term $d(x, c)$ utilizes $L_{\text{intra}}$ and $L_{\text{inter}}$ to regularize the reconstruction distance $d_{\text{rec}}$ as shown in Eq. 4. This distance is used both for cluster assignment and for the logits that supervise $M$.

$$d(x, c) = d_{\text{rec}}(x, c) + w_{\text{intra}} L_{\text{intra}} - w_{\text{inter}} L_{\text{inter}} \tag{4}$$

with weights $w_{\text{intra}}, w_{\text{inter}}$ balancing the pull–push forces and $d_{rec}$ is the reconstruction-based distance between a feature vector $x$ and class $c$ representation. Basically, the regularizer terms shrink intra-class spread and enlarge prototype margins, leading to cleaner pseudo labels and superior few-shot accuracy relative to the previous methods (see Table 5).

In each episode, both support and query samples $x$ are (re)labeled with the class that minimizes the above distance. After that, every prototype is reset to the mean of its current members. Next, we refine the prototypes with a lightweight *cluster separation tuner* (CST) modelled after the continuous firefly heuristic (Yang & He, 2013). At every CST iteration, we compute a *brightness* score $B(c)$ for the prototype of each class c.

$$B(c) \;=\; -w_{\text{intra}} L_{\text{intra}}(c) + \; w_{\text{inter}} L_{\text{inter}}(c) - \sum_{\substack{c' \neq c \\ \|P(c)-P(c')\|_2 < \varepsilon}} \left(\varepsilon - \|P(c) - P(c')\|_2\right)^2 \quad (5)$$

where the first two terms reuse the same intra- and inter-class distances, but taken *class-wise* (the intra-class and inter-class distances are computed separately for each individual class rather than averaged over all classes within the episode) rather than episode-wise here, and the last term penalizes prototypes that sit closer than an $\varepsilon$ margin to any neighbour. If prototype $P(c)$ is dimmer than $P(c')$ ($B(c) < B(c')$) we move it towards the brighter neighbour with an attractiveness

$$\beta = \beta_0 \exp\left(-\gamma \|P(c) - P(c')\|_2^2\right), \quad (6)$$

yielding the update

$$P(c) \;\leftarrow\; P(c) + \beta\left[P(c') - P(c)\right] + \alpha\left(\text{rand}[-\tfrac{1}{2}, \tfrac{1}{2}]\right). \quad (7)$$

Here $\beta_0$ sets the maximum attraction, $\gamma$ controls its spatial decay, and the Gaussian–type exploration term of amplitude $\alpha$ (decayed each loop by $0.995$) prevents premature convergence. Iterating this procedure for a small, fixed number of loops maximizes prototype separation while keeping each class center tethered to its own support cloud. Our method synthesizes the complementary strengths of reconstruction-based and Euclidean-based metrics. The reconstruction residual captures the intrinsic local geometry, while the Euclidean regularizer enforces global separability. The Cluster Separation Tuner (CST) further refines this by explicitly extending decision boundaries to maximize inter-class margins. For additional details, implementation specifics, and related experiments, please refer to Sec. A.3 in the Appendix.

To predict labels for each query embedding $x_q$, we first compute its per-class reconstruction residuals $d_{\text{rec}}(x_q, c)$. These distances are then converted into logits as follows.

$$\ell(x_q, c) \;=\; -\log\left(d_{\text{rec}}(x_q, c) + \varepsilon\right), \quad (8)$$

Only reconstruction residuals are used at inference because the variance optimization terms ($L_{\text{intra}}, L_{\text{inter}}$) function as global structural constraints during the iterative transductive tuning phase to refine the class prototypes and separate clusters. However, the equation just above represents the final inference step *after* these prototypes have converged. At this stage, because the prototypes are already spatially optimized to account for class variance, we rely solely on the reconstruction residual to provide the most precise, instance-level measure of fit.

They are then normalized via softmax to obtain class probabilities as follows.

$$p(c \mid x_q) \;=\; \frac{\exp\left(\ell(x_q, c)\right)}{\sum_{c'} \exp\left(\ell(x_q, c')\right)}, \quad (9)$$

Finally the predicted label is obtained as: $\hat{y}_q \;=\; \arg\max_c p(c \mid x_q)$

### 4.3.2 EPISODIC FINE-TUNING LOSSES

Once CVOC has delivered a set of refined prototypes and the corresponding query logits $Q_{\text{cvoc}} \in \mathbb{R}^{(N_q C) \times C}$, we optimize the visual extractor $M$ with three episode-level losses:

The CVOC logits are compared with the ground-truth query labels through a cross-entropy term

$$L_{\text{cvoc}} = L_{\text{ce}}(Q_{\text{cvoc}}, y_q). \quad (10)$$

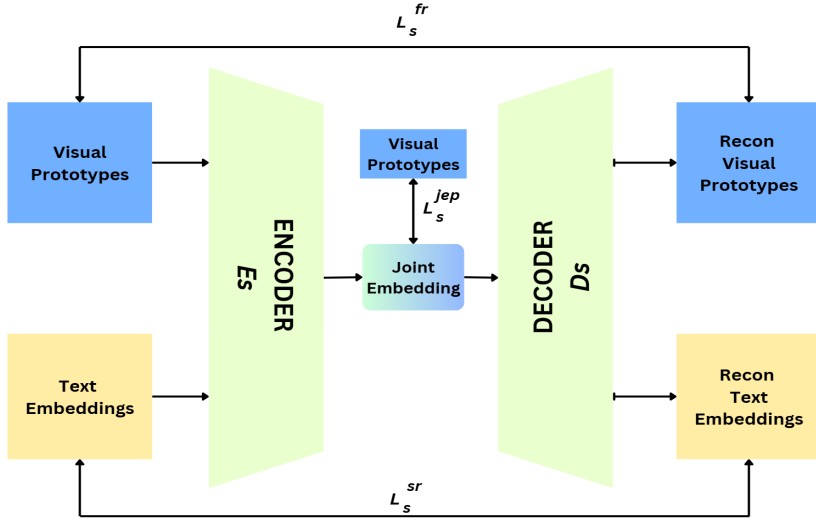

Figure 2: Semantic Injection Network

In parallel, we pass the same query embeddings through the label propagation component (Iscen et al., 2019), producing logits $Q_{\text{lp}}$. The resulting loss is as follows.

$$L_{\text{lp}} = L_{\text{ce}}(Q_{\text{lp}}, y_q). \tag{11}$$

To stabilize the training, we also apply the classifier head to every support and query embedding, yielding the following loss.

$$L_{\text{cls}} = L_{\text{ce}}(Q_{\text{all}}, y_{\text{all}}). \tag{12}$$

The total objective is a convex combination as shown below.

$$\mathcal{L}_{\text{episode}} = w_{\text{cls}} L_{\text{cls}} + w_{\text{fs}}\Big( \eta\, L_{\text{cvoc}} + (1 - \eta)\, L_{\text{lp}} \Big), \tag{13}$$

where $w_{\text{cls}}$ controls the contribution of the auxiliary episode-wide term, $w_{\text{fs}}$ the overall few-shot weight, and $\eta \in [0, 1]$ trades off CVOC and label propagation for the query samples. All weights are fixed on a validation split and kept constant for every episode.

## 4.4 TRAINING SEMANTIC INJECTION NETWORK

Once the pre-training and finetuning of the visual feature extractor $(M)$ are completed, we train a semantic injection network $(S)$. For training this network, we need class descriptions. We propose a simple, automated class description generation pipeline to generate high-quality, visually informative class descriptions (refer to Sec. A.4.4 in the Appendix). This is done to avoid any ambiguities that may arise from using just the class names. We use CLIP (Radford et al., 2021) text encoder to obtain the class embeddings/semantic features from the generated class descriptions. We extract features of images $x_i$ from the train set using the fine-tuned visual feature extractor $M$ and embedding propagation. We obtain the class prototypes $P_{c_j}$ by simply averaging the features from corresponding training samples from each class $c_j$. We concatenate the class prototype with the semantic features of the same class and pass the concatenated it to the encoder module $E_s$ of the semantic injection network to obtain a joint embedding $E_s\big(P_{c_j}, s_{c_j}\big)$. The decoder module $D_s$ of the semantic injection network is used to reconstruct the concatenated feature that was passed into $E_s$.

We train the semantic injection network S using three loss functions: joint embedding prototype loss $L_s^{jep}$, image feature prototype reconstruction loss $L_s^{fr}$, and semantic feature reconstruction loss $L_s^{sr}$ shown in Eq. 14. The joint embedding prototype loss $L_s^{jep}$ helps move the joint embedding closer to the image feature prototype of the same class (see Eq. 15). The image feature prototype of a class is computed by taking the mean of the features of images belonging to that class (see Eq. 16).

Table 1: Classification accuracy (%) on the miniImageNet dataset.

| Method | Venue | ResNet-12 | | WRN-28-10 | |
| --- | --- | --- | --- | --- | --- |
| | | 1-shot | 5-shot | 1-shot | 5-shot |
| TADAM | NeurIPS'18 | 58.50±0.30% | 76.70±0.30% | 61.76±0.08% | 77.59±0.12% |
| CAN | NeurIPS'19 | 67.19±0.55% | 80.64±0.35% | 62.96±0.62% | 78.85±0.10% |
| LST | CVPR'18 | 70.10±1.90% | 78.70±0.80% | 70.74±0.85% | 84.34±0.53% |
| EPNet | ECCV'20 | 66.50±0.89% | 81.06±0.60% | 62.93±1.11% | 82.24±0.59% |
| TPN | ICLR'19 | 59.46% | 75.65% | 71.41% | 81.12% |
| PLAIN | ICME'21 | 74.38±2.06% | 82.02±1.08% | 79.22±0.92% | 88.05±0.51% |
| EPNet-SSL | ECCV'20 | 75.36±1.01% | 84.07±0.60% | 81.57±0.94% | 87.17±0.58% |
| cluster-FSL | CVPR'22 | 77.81±0.81% | 85.55±0.41% | 82.63±0.79% | 89.16±0.35% |
| **Ours** | – | **84.51±0.54%** | **86.95±0.39%** | **85.94±0.54%** | **91.03±0.34%** |

The image feature prototype reconstruction loss $L_s^{fr}$ is used to reduce the difference between the reconstructed prototype $P_{c_j}^r$ and the actual prototype feature $P_{c_j}$. Similarly, the semantic feature reconstruction loss $L_s^{sr}$ is used to reduce the difference between the reconstructed semantic feature $s_{c_j}^r$ and the actual semantic feature $s_{c_j}$. The $L_s^{fr}$ and $L_s^{sr}$ losses force the semantic injection network to incorporate semantic information into the encoded prototype produced by $E_s$ (see Fig. 3).

$$L_s = L_s^{jep} + L_s^{fr} + L_s^{sr} \qquad (14) \qquad\qquad L_s^{jep} = F_{L1}\big(E_s(P_{c_j}, s_{c_j}), P_{c_j}\big) \qquad (15)$$

$$P_{c_j} = \frac{1}{N_{c_j}} \sum_{(x_i, y_i) \in D_{tr}} \mathbb{1}(y_i = c_j) \, \mathrm{ep}\big(M(x_i)\big) \qquad (16)$$

$$P_{c_j}^r, s_{c_j}^r = D_s\big(E_s(P_{c_j}, s_{c_j})\big) \quad (17) \qquad L_s^{fr} = F_{L1}(P_{c_j}^r, P_{c_j}) \quad L_s^{sr} = F_{L1}(s_{c_j}^r, s_{c_j}) \quad (18)$$

Where, $F_{L1}$ refers to L1-loss, $P_{c_j}$ refers to the class $c_j$ prototype, $N_{c_j}$ refers to the number of labeled samples belonging to class $c_j$.

### 4.5 Evaluation with Semantic Injection and Restricted Pseudo-Labeling

After pre-training and episodic fine-tuning are completed and the semantic injection network has been trained, we evaluate the model on $N$-way $K$-shot test episodes composed of a labeled support set $\mathcal{S}_{\text{test}}$, an unlabeled pool $\mathcal{U}_{\text{test}}$, and a query set $\mathcal{Q}_{\text{test}}$. We first run CVOC on $\mathcal{S}_{\text{test}} \cup \mathcal{U}_{\text{test}}$ to obtain the visual centers $P(y)$ for each class. For each class $y$ we encode its textual description with the CLIP text encoder and obtain a semantic vector $t_y$ (Radford et al., 2021). We process the concatenated $P(y)$ and $t_y$ through the encoder $E_s$ of the semantic-injection network yields the refined prototype $P^{\text{ref}}(y)$.

$$P^{\text{ref}}(y) = s\, P(y) + (1 - s)\, E_s\big(P(y), t_y\big), \qquad (19)$$

where $s \in [0, 1]$ is a fixed blend weight. This constitutes the overall procedure of our semantic anchor module — aligning visual prototypes via the semantic injection network and prototype correction according to Eq. 19. Visual features alone are often ambiguous in few-shot settings. The text embedding acts as a noise-free prior that anchors the visual prototype.

For each unlabeled sample $x_i^u \in \mathcal{U}_{\text{test}}$ we compute the entropy $H(x_i^u)$ of its softmax logits and retain only the lowest-entropy $k\%$ samples. The selected subset and its hard pseudo-labels are computed as

$$\widetilde{\mathcal{U}}_{\text{test}} = \Big\{(x_i^u, \tilde{y}_i) \mid H(x_i^u) \le H_k\Big\}, \qquad \tilde{y}_i = \arg\max_c \mathrm{logit}_c(x_i^u), \qquad (20)$$

where $H_k$ is the entropy threshold that keeps exactly $k\%$ of $\mathcal{U}_{\text{test}}$.

The support set is then expanded with these high-confidence examples: $S^0 = \mathcal{S}_{\text{test}} \cup \widetilde{\mathcal{U}}_{\text{test}}$. Balanced label propagation (Iscen et al., 2019) is applied to the features of $S^0 \cup \mathcal{Q}_{\text{test}}$, using the labels in $S^0$ as anchors. The propagated soft labels for $\mathcal{Q}_{\text{test}}$ are finally hardened by selecting the highest-probability class, yielding the episode predictions (Fig. 3).

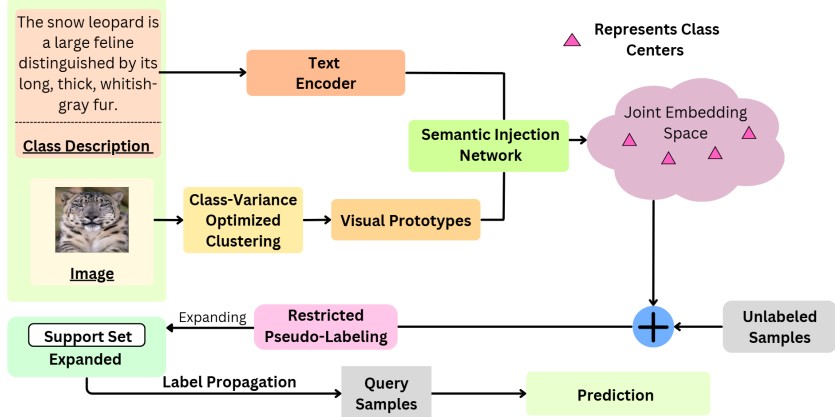

Figure 3: Evaluation using semantic injection and restricted pseudo-labeling

Table 2: Classification accuracy (%) on the tieredImageNet dataset.

| Method | Venue | ResNet-12 | | WRN-28-10 | |
|---|---|---|---|---|---|
| | | 1-shot | 5-shot | 1-shot | 5-shot |
| MetaOpt-SVM | CVPR'19 | 65.99±0.72% | 81.56±0.53% | – | – |
| Soft K-Means | ICLR'18 | – | – | 51.52±0.36% | 70.25±0.31% |
| CAN | NeurIPS'19 | 73.21±0.58% | 84.93±0.38% | – | – |
| LEO | ICLR'19 | – | – | 66.33±0.05% | 81.44±0.09% |
| LST | CVPR'18 | 77.70±1.60% | 85.20±0.80% | – | – |
| wDAE-GNN | CVPR'19 | – | – | 68.16±0.16% | 83.09±0.12% |
| EPNet | ECCV'20 | 76.53±0.87% | 87.32±0.64% | 78.50±0.91% | 88.36±0.57% |
| PLML-EP | IEEE TIP'24 | 79.62% | 86.69% | – | – |
| ICI | CVPR'20 | – | – | 85.44% | 89.12% |
| PLAIN | ICME'21 | 82.91±2.09% | 88.29±1.25% | – | – |
| EPNet-SSL | ECCV'20 | 81.79±0.97% | 88.45±0.61% | 83.68±0.99% | 89.34±0.59% |
| PTN | AAAI'21 | – | – | 84.70±1.14% | 89.14±0.71% |
| cluster-FSL | CVPR'22 | 83.89±0.81% | 89.94±0.46% | 85.74±0.76% | 90.18±0.43% |
| **Ours** | – | **85.55±0.65%** | **91.13±0.31%** | **89.06±0.58%** | **91.53±0.32%** |

## 5 EXPERIMENTS

### 5.1 RESULTS

We conducted experiments under 5-way 1-shot and 5-way 5-shot settings, on the miniImageNet, tieredImageNet, and CUB-200-2011 datasets, using WRN-28-10 and ResNet-12 backbones. For the miniImageNet dataset (Table **??**), our approach outperformed cluster-FSL (Ling et al., 2022), achieving a 6.7% and 1.4% improvement in 1-shot and 5-shot settings with ResNet-12, and a 3.31% and 1.87% improvement with WRN-28-10. On the tieredImageNet dataset (Table **??**), the proposed approach outperforms the closest method by margins of 1.66% and 1.19% for 1-shot and 5-shot cases with ResNet-12, and by a margin of 3.32% for 1-shot and 1.35% for 5-shot settings with WRN-28-10. On the CUB-200-2011 dataset (Table 3), our approach outperforms EPNet (Rodríguez et al., 2020) and cluster-FSL (Ling et al., 2022) by significant margins. All experiments were conducted on 1000 few-shot tasks with 100 unlabeled data, and 95% confidence intervals are reported.

### 5.2 ABLATION EXPERIMENTS

We experimentally analyze the contributions of components of our method on the miniImageNet dataset and ResNet12 architecture. The results in Table 4 indicate that removing the semantic

Table 3: Classification accuracy (%) on the CUB dataset.

| Method | Venue | ResNet-12 | | WRN-28-10 | |
| --- | --- | --- | --- | --- | --- |
| | | 1-shot | 5-shot | 1-shot | 5-shot |
| EPNet | ECCV'20 | 82.85±0.81% | 91.32±0.41% | 87.75±0.70% | 94.03±0.33% |
| cluster-FSL | CVPR'22 | 87.36±0.71% | 92.17±0.31% | 91.80±0.58% | 95.07±0.23% |
| **Ours** | – | **88.63±0.63%** | **93.37±0.15%** | **93.15±0.41%** | **96.25±0.24%** |

Table 4: Impact of the intra-inter class distance in finetuning and semantic injection in the testing phase on miniImageNet.

| Components | | Accuracy | |
| --- | --- | --- | --- |
| Intra–inter class dist | Semantic injection | 5w1s | 5w5s |
| × | × | 78.31±0.79% | 85.75±0.41% |
| ✓ | × | 80.29±0.54% | 85.91±0.79% |
| × | ✓ | 82.53±0.52% | 86.45±0.41% |
| ✓ | ✓ | **84.51±0.54%** | **86.95±0.39%** |

Table 5: Pseudo-labeling accuracy (%) on miniImageNet.

| Method | 5-way 5-shot | | 5-way 1-shot | |
| --- | --- | --- | --- | --- |
| | ResNet-12 | WRN-28-10 | ResNet-12 | WRN-28-10 |
| K-Means | 84.13% | 87.62% | 76.67% | 81.39% |
| MFC | 85.04% | 88.38% | 77.26% | 81.90% |
| **CVOC** | 86.65% | 90.21% | 83.98% | 85.41% |

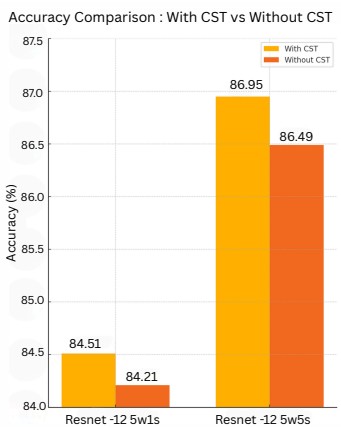

Figure 4: Effect of cluster separation tuner on the miniImageNet dataset 5w1s and 5w5s using ResNet12.

injection, or intra-inter class distance, significantly degrades the performance of our method. Fig. 4 indicates the utility of CST. Please refer to the *Appendix* for other ablation experiments and results.

## 6 CONCLUSION

In this work, we proposed a novel SSFSL approach that performs improved clustering-based pseudo-labeling that promotes better intra/inter-class alignment and leverages semantic anchoring. Extensive experiments demonstrate that our method significantly outperforms recent state-of-the-art approaches.

## 7 REPRODUCIBILITY STATEMENT

We have taken several steps to ensure the reproducibility of our work. The details of our proposed SSFSL framework, including the class variance optimized clustering (CVOC) process and the semantic injection network, are described in Sec. 4. Hyperparameter settings, training procedures, and model configurations are provided in the Appendix Sec. A.3. A complete description of the datasets, and evaluation protocols is also included in the Appendix Sec. A.3. For clarity, the pseudo code of the algorithms used are outlined in the Appendix as well. To facilitate replication of our experiments, we provide anonymous access to the source code and scripts in the anonymous repository link.

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

# A   APPENDIX

## A.1   GENERALIZATION AND ROBUSTNESS EVALUATION

To rigorously evaluate the robustness and generalization capabilities of our proposed method, we conducted a series of experiments under challenging conditions that simulate real-world scenarios. These include performance evaluation in open-set settings with distractor classes and cross-domain few-shot learning tasks. The results underscore the model's ability to maintain high performance beyond standard benchmark conditions.

### A.1.1   PERFORMANCE IN DISTRACTIVE SETTINGS

A more realistic semi-supervised few-shot learning scenario is the distractive semi-supervised few-shot learning setting, where the unlabeled data includes samples from distractor classes that are different from the classes in the episode. Most existing few-shot learning studies do not consider this setting. We evaluated our approach in this setting on the *mini*ImageNet dataset.

As shown in Table 6, our method achieves a new state-of-the-art performance, outperforming previous approaches. This demonstrates the strong open-set recognition capability of our framework, highlighting its accuracy and stability when faced with unknown classes. For this setting, we follow the experimental setup of (Lazarou et al., 2025). The collection of unlabeled data is broadened by incorporating examples from three additional *distractor* classes. These classes are distinct and do not overlap with the actual classes in the episode. For the distractor classes, 30 examples are added per

class in 1-shot epsiodes/tasks and 50 per class in 5-shot episodes/tasks under the 30/50 configuration. This expansion results in a total of 240 unlabeled examples for 1-shot tasks and 400 for 5-shot tasks in this specific setup. The closest baseline iLPC (Lazarou et al., 2025) proposes a synergistic approach of predicting pseudo-labels by leveraging the data manifold and interpreting confident pseudo-label prediction as a form of label cleaning; whereas our approach advances clustering quality, representation learning, and semantic information injection for more accurate pseudo-labeling rather than relying on direct label cleaning from manifold-based confidence.

Table 6: Performance on *mini*ImageNet under the Distractive Semi-Supervised Few-Shot Learning Settings.

| Method | Network | 5-way 1-shot (%) | 5-way 5-shot (%) |
|---|---|---|---|
| Masked soft k-means  (Ren et al., 2018) | ResNet-12 | 61.00 | 72.00 |
| TPN  (Liu et al., 2018) | ResNet-12 | 61.30 | 72.40 |
| LST  (Sung et al., 2018) | ResNet-12 | 64.10 | 77.40 |
| MCT  (Kye et al., 2020) | ResNet-12 | $69.60 \pm 0.70$ | $81.30 \pm 0.50$ |
| iLPC  (Lazarou et al., 2025) | WRN-28-10 | $76.22 \pm 0.86$ | $87.15 \pm 0.43$ |
| **Ours** | **ResNet-12** | **$74.92 \pm 0.71$** | **$85.65 \pm 0.44$** |
| **Ours** | **WRN-28-10** | **$78.59 \pm 0.66$** | **$89.98 \pm 0.36$** |

### A.1.2 GENERALIZATION TO DOMAIN SHIFT

To assess the ability of our approach to generalize to new data distributions, we performed a stringent cross-domain evaluation. The ResNet-12 backbone, pre-trained and fine-tuned only on *mini*ImageNet, was tested on several target datasets with significant domain shifts: CUB, EuroSAT, ISIC, and ChestX. For these experiments, we evaluate performance with the semantic anchor module disabled (to isolate the core visual framework), as well as enabled with training on either the source domain or a subset of the target data.

The results, presented in Table 7, demonstrate strong cross-domain generalization. Our approach coupled with semi-supervised framework is even competitive with and often surpasses methods specifically designed for domain adaptation eg. self-supervised learning (SSL) methods, even without any training on the target domains. This highlights the intrinsic generalization power of our proposed clustering. We follow (Oh et al., 2022) to rank our target datasets. The established order for domain similarity to ImageNet is CUB > EuroSAT > ISIC > ChestX, and for few-shot difficulty is ChestX > ISIC > CUB > EuroSAT.

Based on the results, we can deduce a particular *failure mode* of the semantic anchor module: the module fails to achieve optimal performance when relying solely on source-domain (miniImageNet) training, requiring access to in-domain target data to fully resolve geometric misalignment and generate the highest results.

### A.1.3 QUALITATIVE ANALYSIS VIA CLASS ACTIVATION MAPPING

To offer a qualitative insight into the generalization of the learned representations for our approach, we visualize and compare Class Activation Maps (CAMs) on a target domain unseen during training. Specifically, we evaluate models that were pre-trained and fine-tuned exclusively on *mini*ImageNet, and then generate CAMs using Grad-CAM as proposed by Selvaraju et al. Selvaraju et al. (2017) on images from the CUB dataset. This cross-domain setup provides a stringent test of how well the learned feature space transfers to novel data distributions. We selected Cluster-FSL by Ling et al. (Ling et al., 2022) as our primary baseline due to its methodological similarity since it also employs a clustering-based strategy in the finetuning stage. This makes it an ideal candidate for comparing the generalization quality of our proposed Class-Variance Optimized Clustering (CVOC).

As shown in Figure 5, our approach demonstrates superior cross-domain generalization. On the out-of-domain CUB dataset, its activations are more semantically focused on the target object, unlike the diffuse activations from Cluster-FSL. This indicates that our method promotes a more robust and transferable representation, which directly facilitates more accurate clustering on novel domains and underpins our strong quantitative results.

Table 7: Cross-Domain Few-Shot Classification Accuracy (5-way 1-shot). We report the results for the SSL methods from (Oh et al., 2022).In contrast to SSFSL setup, transductive FSL setup has no additional unlabeled set. SA: Semantic Anchor.

| Method | Network | CUB | EuroSAT | ISIC | ChestX |
|---|---|---|---|---|---|
| *SSL pre-training on target domain* | | | | | |
| SimCLR (Chen et al., 2020) | ResNet-18 | - | 84.30% | 36.39% | 21.55% |
| MoCo (He et al., 2020) | ResNet-18 | - | 69.11% | 29.54% | 21.74% |
| BYOL (Grill et al., 2020) | ResNet-18 | - | 66.16% | 34.53% | 22.75% |
| SimSiam (Chen & He, 2021) | ResNet-18 | - | 70.80% | 30.17% | 22.17% |
| *No target domain training; trained on miniImageNet; Transductive FSL inference* | | | | | |
| RDC (Li et al., 2022) | ResNet-10 | 47.77% | 67.58% | 32.29% | 22.66% |
| LDP-net (Zhou et al., 2023) | ResNet-10 | 55.94% | 73.25% | 33.44% | 22.21% |
| *No target domain training; trained on miniImageNet; SSFSL inference; without SIN* | | | | | |
| Cluster-FSL (Ling et al., 2022) | ResNet-12 | 56.83% | 70.78% | 33.72% | 21.81% |
| **Ours** | ResNet-12 | **60.95%** | **75.66%** | **36.25%** | **21.90%** |
| *No target domain training; trained on miniImageNet; SSFSL inference; with SA trained on miniImageNet* | | | | | |
| **Ours** | ResNet-12 | **61.45%** | **75.86%** | **36.20%** | **21.73%** |
| *No target domain training; trained on miniImageNet; SSFSL inference; with SA trained on subset of target data not in test target data* | | | | | |
| **Ours** | ResNet-12 | **62.75%** | **77.36%** | **37.10%** | **22.42%** |

## A.2 NOTATION TABLE

The notations used throughout this paper are summarized in Table 8.

## A.3 EXPERIMENTAL DETAILS

### A.3.1 DATASETS

The *mini*ImageNet dataset introduced by Vinyals et al. (2016) is a subset of ImageNet by Russakovsky et al. (2015), consisting of 60,000 images from 64 training classes, 16 validation classes, and 20 testing classes. The *tiered*ImageNet dataset proposed by Ren et al. (2018) is another ImageNet subset designed specifically for few-shot learning applications. This dataset of 779,165 images includes a hierarchical structure with 34 superclasses divided into 608 fine-grained categories, encompassing a wide range of natural and artificial objects, animal species, and other general image classes. It is partitioned into 20 training superclasses (351 classes), 6 validation superclasses (97 classes), and 8 test superclasses (160 classes). The CUB dataset introduced by Hilliard et al. (2018), derived from the fine-grained Caltech-UCSD Birds-200-2011 dataset by Welinder et al. (2010), contains 200 classes representing 11,788 bird images. It is partitioned into 100, 50, and 50 classes for the base, validation, and novel splits.

For the domain adaptation part, we used the following datasets along with the CUB dataset (Hilliard et al., 2018) -

- **EuroSAT** (Helber et al., 2019) is a set of satellite images of landscapes.

- **ISIC** (Codella et al., 2019) is a set of dermoscopy images of human skin lesions.

- **ChestX** (Wang et al., 2017) is a set of X-Ray images of the human chest.

Our dataset compilations and splits follow those used by (Oh et al., 2022). Figure 6 shows several examples from the target domain datasets.

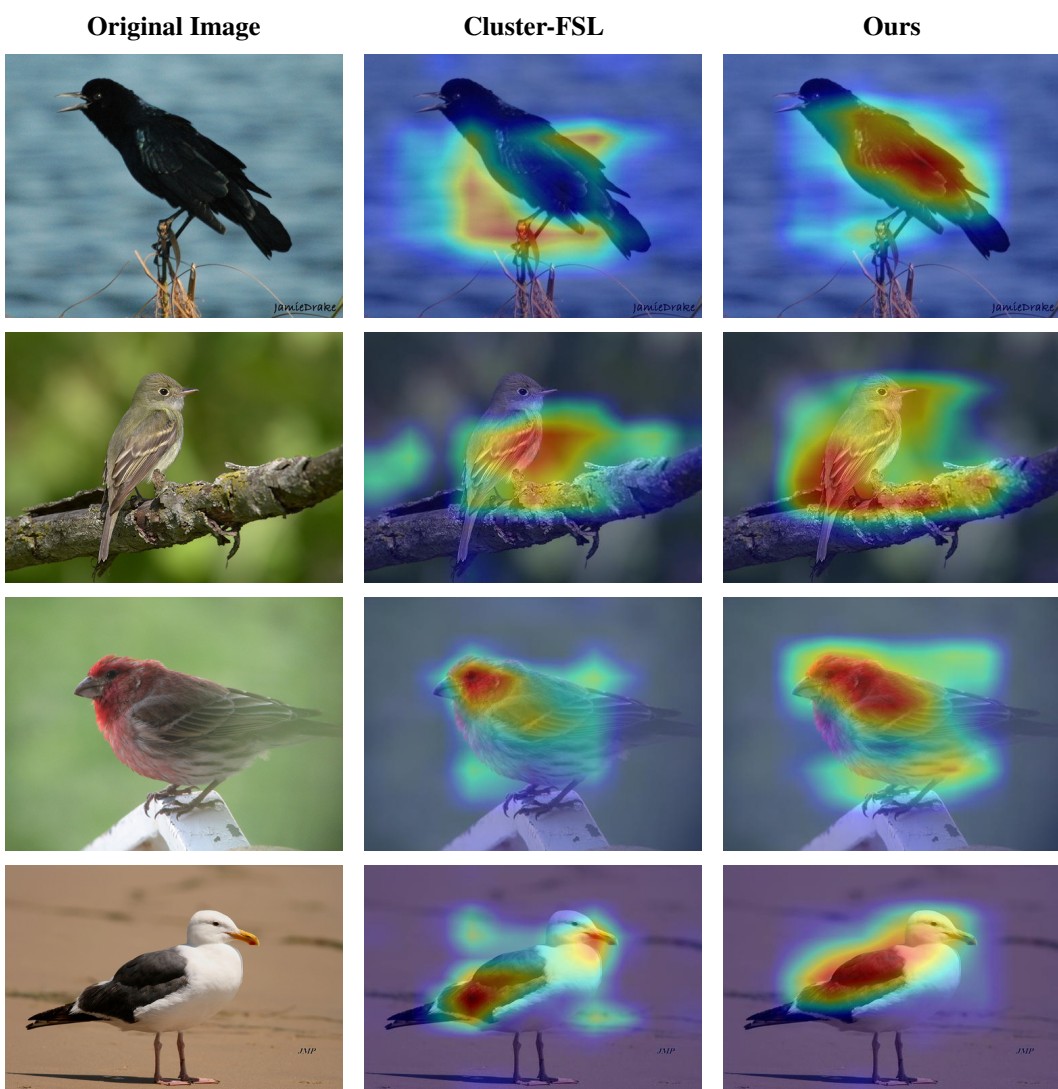

| **Original Image** | **Cluster-FSL** | **Ours** |

Figure 5: Qualitative comparison of cross-domain activation heatmaps on the CUB dataset. All models were trained on *mini*ImageNet. The columns show the original image, the heatmap from Cluster-FSL (often diffuse), and the heatmap from our approach (superior localization), respectively.

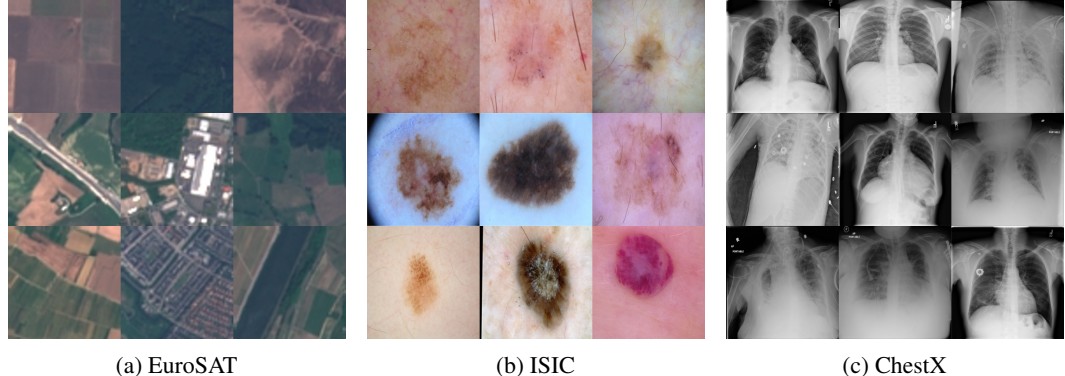

(a) EuroSAT        (b) ISIC        (c) ChestX

Figure 6: Examples from our target domain datasets for cross-domain experiments.

Table 8: List of notations used in this paper.

| Symbol | Description |
|---|---|
| $C_b, C_f$ | Base classes and novel/few-shot classes. |
| $D_{tr}, D_{va}, D_{tt}$ | Training, validation, and test datasets. |
| $N$ | Number of classes in an episode ("N-way"). |
| $K$ | Number of support samples per class ("K-shot"). |
| $S = \{(x_i^s, y_i^s)\}_{i=1}^{N \times K}$ | Support set (labeled examples). |
| $Q = \{(x_i^q, y_i^q)\}_{i=1}^{N \times T}$ | Query set (test samples). |
| $U = \{x_i^u\}_{i=1}^{N \times u}$ | Unlabeled samples in an episode. |
| $M$ | Visual feature extractor network. |
| $S$ (network) | Semantic injection network with encoder $E_s$ and decoder $D_s$. |
| $s_k^{(c)}$ | $k$-th support embedding of class $c$. |
| $P_0^{(c)}$ | Initial prototype of class $c$ (mean of its support embeddings). |
| $P(c)$ | Current prototype of class $c$ (updated during clustering). |
| $\beta_c^*(x)$ | Optimal reconstruction weights of feature $x$ with respect to class $c$. |
| $d_{\text{rec}}(x, c)$ | Reconstruction distance of feature $x$ w.r.t. class $c$. |
| $ep(M(x))$ | Embedding-propagated feature of $x$. |
| $L_{\text{intra}}$ | Intra-class distance loss. |
| $L_{\text{inter}}$ | Inter-class distance loss. |
| $d(x, c)$ | Combined distance (reconstruction + intra/inter regularization). |
| $w_{\text{intra}}, w_{\text{inter}}$ | Weights for intra-class and inter-class regularizers. |
| $\epsilon$ | Small positive constant (numerical stability, prototype margin). |
| $\tau$ | Softmax temperature parameter. |
| $B(c)$ | Brightness score of prototype $c$ in the Cluster Separation Tuner (CST). |
| $\beta, \beta_0, \gamma, \alpha$ | CST parameters: attractiveness, maximum attraction, spatial decay, and random exploration amplitude. |
| $L_{cvoc}$ | Cross-entropy loss with CVOC logits. |
| $L_{lp}$ | Cross-entropy loss with label propagation logits. |
| $L_{cls}$ | Cross-entropy classification loss on all embeddings. |
| $w_{cls}, w_{fs}$ | Loss weights in episodic objective. |
| $\eta$ | Weight balancing CVOC and label propagation in episodic loss. |
| $L_{episode}$ | Final episodic loss combining all terms. |
| $L_s$ | Total loss for training semantic injection network. |
| $L_s^{jep}, L_s^{fr}, L_s^{sr}$ | Semantic injection losses: joint embedding prototype, feature reconstruction, and semantic reconstruction. |
| $P_c^r, s_c^r$ | Reconstructed prototype and semantic feature for class $c$. |
| $P^{ref}(y)$ | Refined prototype of class $y$ via semantic injection. |
| $s$ | Mixing weight between visual prototype and semantic embedding in semantic injection. |
| $H(x)$ | Entropy of logits for unlabeled sample $x$, used in restricted pseudo-labeling. |

### A.3.2 IMPLEMENTATION DETAILS

**Pretraining** During the pre-training stage in our approach, the model is trained for 200 epochs using a batch size of 128, utilizing all available training categories and data. A dropout rate of 0.1, weight decay of 0.0005, and momentum of 0.9 are applied. The model is optimized using the stochastic gradient descent (SGD) algorithm with an initial learning rate set to 0.1. If the validation

loss does not improve for 10 consecutive epochs, the learning rate is reduced by a factor of 10. The output dimensionality of the feature representations depends on the backbone: it is 512 for ResNet-12 and 640 for WRN-28-10.

**Finetuning**  In the fine-tuning stage in our approach, the model is trained for 200 epochs over 600 iterations. The optimization employs stochastic gradient descent (SGD) with a momentum of 0.9 and a weight decay of 0.0005. The learning rate is initially set to 0.001, and it is reduced by a factor of 10 when the validation performance plateaus. Episodic training is done at this stage. To keep the additional computational cost minimal and to ensure that CST (see Alg. 2) does not dominate the main CVOC (see Alg. 1) algorithm, we restrict CST to only one iteration. Meanwhile, the number of iterations in CVOC is set to 10 in all settings. The remaining CST hyperparameters used are $\epsilon = 2.0$, $\beta_0 = 0.05$, $\gamma = 0.005$, $\alpha = 0.02$ across every settings and datasets. This conservative choice of CST update parameters ensures that the overall CVOC algorithm, when coupled with CST, remains stable and efficient, making it well-suited for practical deployment scenarios while being robust and not overly sensitive to a high number of hyperparameters. Refer to A.5.3 to find various ablation studies with $w_{\text{intra}}$ and $w_{\text{inter}}$ and other CST hyperparameters.

**Semantic Injection Network Training**  The Semantic Injection Network in our approach is trained separately after episodic fine-tuning. We train for 100 epochs with a batch size of 128 support–query pairs drawn from all training classes. The input–output dimensions are 512 for image features when using ResNet-12 and 640 when using WRN-28-10 as the backbone. In both cases, the text features (semantic embeddings from CLIP)( Radford et al. (2021)) have a dimensionality of 512. A hidden layer of size 4096 is used. Training is performed using the AdamW optimizer with an initial learning rate of $1 \times 10^{-4}$, a weight decay of $1 \times 10^{-4}$, and a dropout rate of 0.1. The learning rate is decayed by a factor of 0.1 every 30 epochs. The total loss is the mean of three L1 components: prototype reconstruction loss, semantic feature reconstruction loss, and joint embedding prototype loss, as detailed in the main text. The checkpoint with the lowest validation reconstruction loss is selected for downstream few-shot evaluation.

**Testing**  During the testing phase, the model is evaluated over 1000 episodes, and the mean classification accuracy across these episodes is reported as the final performance metric. Each few-shot task consists of 5 classes (N = 5). For every class, there are K = 1 and K = 5 support samples, along with 15 query samples (q = 15) per class. For each category, the number of unlabeled samples $u$ is 100. The temperature parameter $\tau$ is fixed at $0.1$. The clustering process is performed over 10 iterations.

All the hyperparameters were selected through extensive tuning across all three stages: pretraining, finetuning, and testing.

**Compute Resources**  All experiments were conducted on a single NVIDIA A6000 GPU. The pretraining phase utilized up to 30 GB of GPU memory, while the subsequent fine-tuning and testing stages required approximately 3–6 GB, which is comparable to previously existing methods.

### A.3.3 COMPARED METHODS

Our comparative experiments include several established baselines as well as our proposed approach. These include cluster-FSL (Ling et al., 2022), PLML+EP (Dong et al., 2024), EPNet (Rodríguez et al., 2020), and ICI (Wang et al., 2020), along with state-of-the-art few-shot learning methods such as TADAM (Oreshkin et al., 2018), MTL (Sun et al., 2019), MetaOpt-SVM (Lee et al., 2019), CAN (Hou et al., 2019), LST (Sung et al., 2018), and LEO (Rusu et al., 2018).

We also evaluate against semi-supervised techniques like TPN (Liu et al., 2018), TransMatch (Yu et al., 2020), and PLAIN (Li et al., 2021), as well as graph-based models such as wDAE-GNN (Gidaris & Komodakis, 2019). Furthermore, comparisons are made with PTN (Huang et al., 2021a) and clustering-based methods introduced by Ren et al. (2018), including Soft K-Means, Soft K-Means+Cluster, and Masked Soft K-Means, iLPC (Lazarou et al., 2025).

Unlike cluster-FSL, we use both inter-class and intra-class variance, and unlabeled samples with low prediction entropy to enhance testing performance.

## A.4 MORE DETAILS ON CVOC

### A.4.1 BACKGROUND ON PROPAGATION METHODS AND INFERENCE PARADIGMS

In semi-supervised few-shot learning (SSFSL), leveraging unlabeled data in addition to very few labeled support examples is a promising way to improve generalization. Two graph-based propagation techniques that have been used in prior works are **Embedding Propagation (EP)** and **Label Propagation (LP)**. These techniques are especially effective in **transductive** settings, where the model sees unlabeled query points during adaptation. Below we briefly review their principles, how they have been applied in SSFSL, and then state our use of them.

**Embedding Propagation (EP)**   Embedding Propagation was introduced by Rodríguez et al. (2020), which enforces smoothness over the embedding space by interpolating embeddings along a similarity graph. After extracting features for all samples, it constructs a similarity graph and propagates embeddings through this graph to obtain smoothed embeddings. This smoothing helps reduce noise and makes decision boundaries more robust, which is particularly useful in semi-supervised or transductive few-shot settings.

**Label Propagation (LP)**   Label Propagation was introduced by Zhu & Ghahramani (2002) as a classical method in which a small number of nodes have known labels, and labels are diffused through a graph over unlabeled nodes. In few-shot and SSFSL domains, LP has been widely used to propagate label information from support examples to query or unlabeled points, improving classification performance in low-label regimes. LP has been adopted in several notable SSFSL methods, including Liu et al. (2018) and other graph-based SSFSL approaches that refine or expand label information through unlabeled samples.

**Usage in SSFSL and Prior Works**   Across the SSFSL literature, EP and LP have become standard building blocks. EP has been used to regularize embedding spaces, while LP has been employed to propagate supervision to unlabeled data. Methods such as EPNet, TPN, and related propagation-based approaches highlight the central role of these techniques in advancing SSFSL.

**Our Usage**   In our method, we also adopt both EP and LP. We employ EP to smooth embeddings over the graph of support, query, and unlabeled data, and then apply LP to infer labels for unlabeled and query samples. This design choice aligns our approach with prior SSFSL works while contributing to improved generalization.

### A.4.2 ALGORITHMIC DETAILS

**Finetuning**   In the fine-tuning stage, we invoke Alg. 1, which in each episode computes the reconstruction distance of all support and query embeddings against class-specific dictionaries, applies intra- and inter-class distance regularization and a Cluster Separation Tuner (CST, see Alg. 2) to refine the prototypes, and finally derives query logits (soft labels) from the updated prototypes. Experimentally, CST improves classification accuracy, but introduces additional computational overhead: the algorithm with CST takes approximately 10-11 seconds per epoch on average, whereas without CST it takes only 8-9 seconds per epoch. We found that CST works best with low iterations (1-3). Higher iterations (5-10) can cause prototypes to drift too far, leading to instability, and also increase the training time too much (with 10 iterations, training time per epoch becomes 1.8x). Also, the "1 iteration" choice is a deliberate design for Efficiency. Since the prototypes are already initialized near the true mean by CVOC, a single CST step acts as a fine-grained "boundary sharpening" operation rather than a global optimization, ensuring stability and minimal computational cost.

**Testing**   At the inference time, we first apply CVOC to the combined support and unlabeled samples to obtain a subset of high-confidence samples and their pseudo-labels. These pseudo-labels are merged into the episode dictionary, and the updated combined support, unlabeled, and query embeddings are passed to a Label Propagation module (Iscen et al., 2019), which produces query logits and final predictions.

The semantic anchor module significantly improves accuracy, while only slightly increasing the inference time, as can be seen in Table 9. For this module, we also introduce an extra hyperparameter that

---

**Algorithm 1** Class–Variance–Optimized Clustering (CVOC) in the finetuning stage

---

**Require:** labelled support embeddings $S = \{(\mathbf{s}_i, y_i)\}_{i=1}^{N_s}$, query embeddings $Q = \{\mathbf{q}_j\}_{j=1}^{N_q}$, intra/inter weights $w_{\text{intra}}$, $w_{\text{inter}}$, ridge coefficient $\lambda$, maximum loops $T$, small constant $\varepsilon > 0$, softmax temperature $\tau$
**Ensure:** query logits $Q_{\text{cvoc}} \in \mathbb{R}^{N_q \times C}$ and hard labels $\hat{y}$
1: Initialise prototype of each class $c$ as the mean of its support: $\mu_c \leftarrow \frac{1}{|S_c|} \sum_{\mathbf{s}_i \in S_c} \mathbf{s}_i$
2: $A \leftarrow S \cup Q$          $\triangleright$ concatenate support and query sets
3: **for** $t = 1$ **to** $T$ **do**          $\triangleright$ outer refinement loop
4:      **for** $c = 1$ **to** $C$ **do**
5:          $\mathbf{F}_c \leftarrow \left[\mathbf{s}_1^{(c)}, \ldots, \mathbf{s}_{|S_c|}^{(c)}, \mu_c\right]$
6:      **end for**
7:      **// Calculate Reconstruction Distance**
8:      **for all** $\mathbf{a} \in A$ **do**
9:          $d_{\text{rec}}(a, c) \leftarrow$ resulting reconstruction distance
10:     **end for**
11:     $L_{\text{intra}} \leftarrow$ mean distance of each support to its own prototype
12:     $L_{\text{inter}} \leftarrow$ mean distance of each support to all other prototypes
13:     **// Prototype update**
14:     **for all** $\mathbf{a} \in A$ **do**
15:        $d(a, c) \leftarrow d_{\text{rec}}(a, c) + w_{\text{intra}} L_{\text{intra}} - w_{\text{inter}} L_{\text{inter}}$
16:     **end for**
17:     Assign $A_{\text{lbl}}(\mathbf{a}) \leftarrow \arg\min_c d(a, c)$
18:     **for** $c = 1$ **to** $C$ **do**
19:        $\mu_c \leftarrow \frac{1}{|A_c|} \sum_{\mathbf{a} \in A_c} \mathbf{a}$          $\triangleright \triangleright$ mean of current members
20:     **end for**
21:     $\{\mu_c\} \leftarrow \text{CLUSTERSEPARATIONTUNER}(\{\mu_c\})$          $\triangleright \triangleright$ margin polishing
22:     **if** prototypes converge **then**
23:        **break**
24:     **end if**
25: **end for**
26: **// Final logits for queries**
27: **for all** $\mathbf{q}_j \in Q$ **do**
28:     Recalculate $d_{\text{rec}}(\mathbf{q}_j, c)$ with updated prototypes
29:     $\ell(\mathbf{q}_j, c) \leftarrow -\log\big(d_{\text{rec}}(\mathbf{q}_j, c) + \varepsilon\big)$
30: **end for**
31: $Q_{\text{cvoc}} \leftarrow \text{softmax}(\ell/\tau), \quad \hat{y} \leftarrow \arg\max_c Q_{\text{cvoc}}$
32: **return** $Q_{\text{cvoc}}, \hat{y}$

---

controls the mixing between the visual prototype and its semantic reconstruction (See Section A.5.5). Before inference, we include a separate step to train the *Semantic Injection Network*, which takes approximately 10-12 seconds per epoch on average.

Table 9: Inference time per few-shot episode containing a total of 75 query samples on ResNet-12 (miniImageNet) measured on an NVIDIA A6000 GPU.

| Method | Inference Time | |
|---|---|---|
| | 5w1s | 5w5s |
| Cluster-FSL Ling et al. (2022) | 0.178 s | 0.201 s |
| Without semantic anchor | 0.181 s | 0.204 s |
| With semantic anchor | 0.187 s | 0.208 s |

### A.4.3 WHY LABEL PROPAGATION

We experimentally demonstrate that integrating both CVOC and label propagation during the testing phase leads to a greater improvement in accuracy compared to using label propagation alone (Table 10). This is because CVOC relies heavily on labeled samples. However, the pseudo-labels obtained from the expanded support set may be noisy, which can adversely affect the performance of

---

**Algorithm 2** Cluster Separation Tuner (CST)

---

**Require:** initial prototypes $\{P(c)\}_{c=1}^{C}$, support embeddings $S$ with labels $y$, weights $w_{\text{intra}}, w_{\text{inter}}$, penalty margin $\varepsilon$, parameters $\beta_0, \gamma, \alpha$, iterations $T$
**Ensure:** refined prototypes $\{P(c)\}$
1: **for** $t = 1$ to $T$ **do**
2:     **// Brightness evaluation**
3:     **for** $c = 1$ to $C$ **do**
4:         $\mathcal{S}_c \leftarrow \{\mathbf{s} \in S \mid y(\mathbf{s}) = c\}$
5:         **if** $\mathcal{S}_c = \varnothing$ **then**
6:             $B_c \leftarrow -10^6$              $\triangleright \triangleright$ empty classes stay dim
7:         **else**
8:             $L_{\text{intra}}(c) \leftarrow$ mean squared distance between $P(c)$ and samples in $\mathcal{S}_c$
9:             $L_{\text{inter}}(c) \leftarrow$ mean squared distance between $\mathcal{S}_c$ and all other prototypes
10:        penalty $\leftarrow$ quadratic cost if any prototype lies closer than $\varepsilon$ to $P(c)$
11:        $B_c \leftarrow - w_{\text{intra}} \cdot L_{\text{intra}}(c) + w_{\text{inter}} \cdot L_{\text{inter}}(c) - $ penalty
12:         **end if**
13:     **end for**
14:     **// Update step**
15:     **for** $i = 1$ to $C$ **do**
16:         **for** $j = 1$ to $C$ **do**
17:             **if** $B_i < B_j$ **then**             $\triangleright \triangleright$ prototype $i$ is dimmer
18:                $\beta \leftarrow \beta_0 \exp\!\left(-\gamma \|P(i) - P(j)\|_2^2\right)$
19:                $P(i) \leftarrow P(i) + \beta\left[P(j) - P(i)\right] + \alpha\left(\text{rand}[-\tfrac{1}{2}, \tfrac{1}{2}]\right)$
20:             **end if**
21:         **end for**
22:     **end for**
23:     $\alpha \leftarrow 0.995\,\alpha$             $\triangleright \triangleright$ anneal random exploration
24: **end for**
25: **return** $\{P(c)\}$

---

CVOC. Hence, employing label propagation to predict the labels of query samples during the testing phase is both a practical and effective approach.

Table 10: Comparison of classification accuracy (%) and pseudo-labeling accuracy (%) between LP (Label Propagation) and CVOC+LP methods using the ResNet-12 backbone on the miniImageNet dataset. SSFSL denotes semi-supervised few-shot learning.

| Method | SSFSL Acc. | | Pseudo Labeling Acc. | |
|---|---|---|---|---|
| | **5w1s** | **5w5s** | **5w1s** | **5w5s** |
| LP | 75.51 | 84.83 | 63.42 | 80.45 |
| CVOC + LP | 84.51 | 86.95 | 83.98 | 86.65 |

### A.4.4 CLASS DESCRIPTION GENERATION PIPELINE

To supply the *semantic-anchor* module with concise yet visually informative text, we automatically build a short description for every class name[1] by running a four–stage sequential agentic chain (see Figure 7) using the GPT-4O-MINI model accessed via the *OpenAI API* (OpenAI, 2024). Each stage receives the output of the previous one and is instructed with a fixed prompt; no manual intervention is required.

1. **Writer agent** — rewrites the raw WordNet definition so that it is scientifically correct, free of jargon, and $\leq 150$ words.

2. **Filter agent** — removes any behavioural, functional, or habitat information, keeping only appearance–related cues ($\leq 100$ words).

---

[1] WordNet glosses for each classes are automatically retrieved via the NLTK (Bird et al., 2009) interface to WordNet (Fellbaum, 1998).

3. **Visualizer agent** — further sharpens size, shape, colour, texture, and pattern details, padding only when such cues are commonly known.

4. **Finalizer agent** — polishes the text into a single 50–70-word paragraph that provides a rich visual description of the particular class.

---

**Writer agent prompt**

Please rewrite the '{definition}' for '{class_name}', ensuring scientific accuracy and clarity, while maintaining brevity. The definition should be detailed enough to capture essential characteristics, yet concise, within 150 words. Focus on providing clear and visually identifiable traits of the object. If the original definition includes overly technical or complex terms that aren't needed to describe how the object looks, simplify them into plain, easy-to-understand language. Avoid adding unnecessary or unrelated information, and do not make up or guess details.

---

**Filter agent prompt**

You are the filter agent. Your task is to review the following expanded definition of '{class_name}'. Focus on preserving only the information that directly describes the object's visual traits and remove behaviors, habitats, functions, or technical jargon unrelated to appearance.
Definition: {rewritten_definition}
Please return the refined definition within 100 words and do not hallucinate any information.

---

**Visualizer agent prompt**

You are the visualizer agent. Your task is to improve the following definition by emphasizing the key visual traits of '{class_name}'. Focus on attributes such as size, shape, color, texture, and patterns. If the definition is too brief or incomplete, add further relevant visual attributes based on common knowledge of the class—but do not hallucinate any information. Ensure the result is concise and highlights the most important physical characteristics.
Filtered Definition: {filtered_definition}
Please return a refined definition of no more than 100 words.

---

**Finalizer agent prompt**

You are the finalizer agent. Your task is to take the following visual-focused definition of '{class_name}' and polish it into a single, well-structured paragraph of 50–75 words. Make it succinct and scientifically accurate. If any details conflict, keep only the most relevant, visually descriptive information. Do not hallucinate or invent any new facts.
Visual-Focused Definition: {visual_focused_definition}
Return: A polished, 50–70-word draft-like paragraph emphasizing the key visual traits.

---

This fully automated pipeline yields compact, visually rich class descriptions that consistently improve semantic anchoring. We evaluated several prompt strategies - including a single-shot prompt, the unedited WordNet gloss, alternative agentic chains, and found that the four-stage pipeline described above consistently produced the strongest downstream performance.

Table 11: Ablation Study on Description Generation Strategy for Semantic Anchoring. We provide accuracy on the miniImageNet dataset for the 5-way 1-shot task, using ViT-B/32 CLIP encoding and a ResNet-12 classifier backbone.

| Description Strategy | Description Source (miniImageNet 5-way 1-shot) | Accuracy (%) |
|---|---|---|
| Baseline | Class Name Only | 78.33 |
| Raw Text Input | Unedited WordNet Gloss | 80.81 |
| LLM Baseline | Single-Shot Prompt | 82.55 |
| **Our Proposed Method** | **Agentic Chain (4-Stage Pipeline)** | **84.51** |

**Effectiveness of our approach**

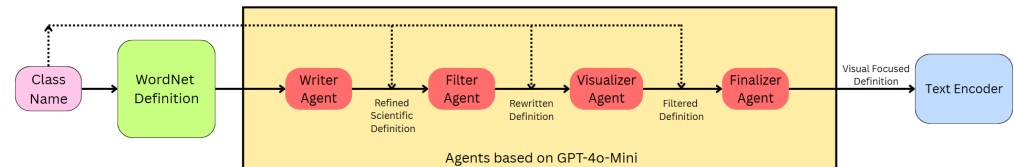

Figure 7: Pipeline for generating visually focused class descriptions using an agentic chain powered by GPT-4o-Mini. Starting from a raw class name, a WordNet definition is retrieved and passed sequentially through four specialized agents: the Writer Agent generates a Refined Scientific Definition; the Filter Agent removes non-visual information; the Visualizer Agent enhances appearance-related cues; and the Finalizer Agent polishes the output into a concise, visually focused description. This final text is then encoded using a Text Encoder for semantic injection. Note that at each step, the class name is passed along with the definition from the previous agent.

Table 12: Semi-supervised few-shot classification on miniImageNet dataset (5-way 1-shot, ResNet-12 backbone). "Num_U" represents the number of unlabeled samples.

| Num_U | Acc (Ours) | Cluster-FSL (Ling et al., 2022) |
|---|---|---|
| 20 | 80.03 | 72.58 |
| 50 | 82.81 | 74.83 |
| 100 | 84.51 | 77.81 |
| 200 | 85.30 | 78.31 |

WHY BEGIN FROM WORDNET. Passing only a bare class name into the writer agent can lead to confusion, because many single words have more than one meaning. For example, the word "crane" could refer to a long-legged bird or a construction machine. WordNet assigns each meaning a unique synset ID and supplies a short gloss for each sense, so we can pick the one needed for the purpose. Starting from these curated senses prevents the large language model (LLM) from mixing unrelated meanings and ensures that every class name is grounded in the correct visual category.

WHY CALL THE LLM. Unfortunately, WordNet glosses are short (often $< 20$ words) and rarely mention fine-grained cues such as colour patches, plumage texture, or limb shape that are crucial for few-shot recognition. We therefore prompt GPT-4O-MINI to *expand* each gloss, injecting the missing visual details.

WHY AN AGENTIC REFINEMENT CHAIN. Iterative self-refinement has been shown to improve factuality and task performance in LLMs (Madaan et al., 2023). So, our four-stage pipeline writer → Filter → Visualizer → Finalizer realises this idea: the *Filter* runs at low temperature to deterministically prune non-visual content, while the *Visualizer* is sampled with a slightly higher temperature to fill in plausible but commonly known traits; the Finalizer then consolidates the description into 50–70 highly informative words. Controlling generation temperature is a well-studied way of trading off creativity and determinism in LLMs.

REMOVING NOISE WITHOUT LOSING SIGNAL. A small amount of non-visual info can help steer the text embedding away from confusion. Aggressive pruning eliminates habitat or behavioural clauses that could hurt visual grounding (e.g. "feeds on carrion") yet keeps concise non-visual hints that implicitly map to appearance: *nocturnal* → large eyes, *aquatic* → streamlined body, *raptor* → hooked beak. Such implicit cues have proved beneficial when text embeddings are later aligned with vision features .

DISAMBIGUATING LOOK-ALIKE CLASSES. Some birds look almost the same in pictures - take an 'American Crow' and 'Common Raven'. Both are shiny black corvids, but ravens are noticeably bigger and have a thicker bill. Standard WordNet definitions do not point out these visual differences, but our process keeps them. That way, our approach learns to differentiate between the two, rather than mixing them up.

Table 13: Semi-supervised few-shot classification on miniImageNet dataset (5-way 5-shot, ResNet-12 backbone)."Num_U" represents the number of unlabeled samples.

| Num_U | Acc (Ours) | Cluster-FSL Ling et al. (2022) |
|---|---|---|
| 20 | 84.94 | 82.89 |
| 50 | 85.89 | 84.39 |
| 100 | 86.95 | 85.55 |
| 200 | 87.43 | 86.25 |

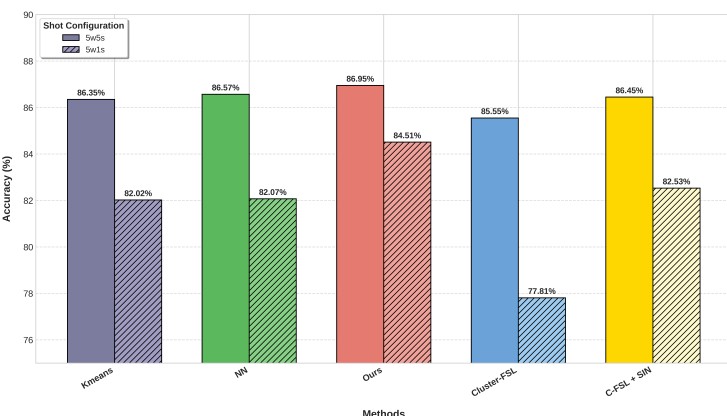

Figure 8: Accuracy comparison across different methods on the miniImageNet dataset using the ResNet-12 backbone for 5-way 1-shot and 5-way 5-shot scenarios. NN: nearest neighbour approach. SIN: Semantic Injection. C-FSL: Cluster-FSL.

This iterative, refinement-driven expansion and denoising procedure yields semantically rich yet compact class texts that translate into stronger visual anchoring and ultimately, higher SSFSL (Semi-Supervised Few-Shot Learning) accuracy.

AGENTIC CHAIN SENSITIVITY ANALYSIS   To further validate the robustness of our agentic architecture, we conducted a sensitivity analysis on various CLIP models and LLMs within our framework. Our primary experiments were conducted using the ViT-B/32 CLIP model. However, as demonstrated in Figure 10b, we observe remarkably similar performance across different ViT architectures, including ViT-B/32, ViT-B/16, and ViT-L/14. This suggests that the benefits of our agentic chain are not solely dependent on the most powerful visual encoder, as even with lower-version CLIP models, we achieve comparable results. Specifically, on the miniImageNet dataset with a 5-way 1-shot task using a ResNet-12 backbone, we obtained an accuracy of 84.51% with ViT-B/32, 84.52% with ViT-B/16, and 86.63% with ViT-L/14. This indicates that our architecture effectively leverages the visual information regardless of the precise CLIP model employed.

Furthermore, we investigated the impact of different LLMs on the performance of our agentic chain. As shown in Figure 10a, our architecture demonstrates consistent strong performance even when integrating weaker LLMs. We hypothesize that the agentic nature of our framework, which facilitates iterative refinement and contextual reasoning, allows for the generation of superior visual description-based texts, thereby mitigating the performance limitations typically associated with less powerful language models. For instance, with GPT-4o-mini, we achieved an accuracy of 84.51%, closely followed by LLama 3.1 8B (Touvron et al., 2023) at 84.31%, Qwen 2.5 7B (Bai et al., 2024) at 84.6%, and Qwen 2.5 3B (Bai et al., 2024) at 84.40%. These results underscore the potential of our agentic architecture to democratize access to high-quality visual understanding, enabling effective performance even with computationally lighter LLMs. All LLM experiments were also conducted on the miniImageNet 5-way 1-shot task with a ResNet-12 backbone.

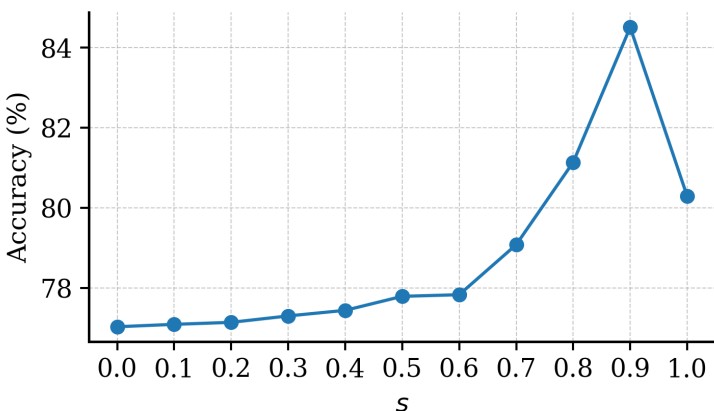

Figure 9: Impact of $s$ on 5-way 1-shot accuracy on the miniImageNet dataset using the ResNet-12 backbone

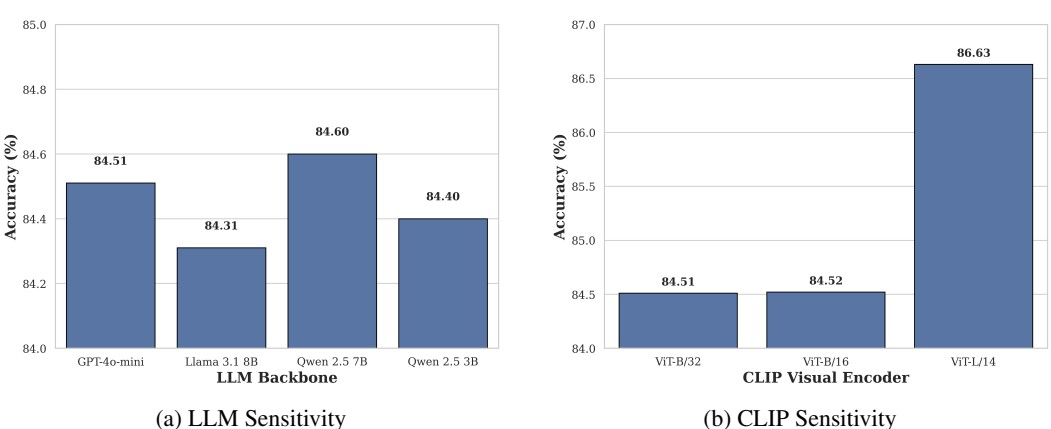

(a) LLM Sensitivity                          (b) CLIP Sensitivity

Figure 10: **Sensitivity Analysis of the Agentic Architecture.** (a) Performance variations across different Large Language Models (LLMs). The plot demonstrates that our agentic framework maintains robust accuracy even when utilizing smaller, computationally efficient models (e.g., Qwen 2.5 3B), supporting the hypothesis that iterative agentic reasoning compensates for lower raw LLM capability. (b) Performance variations across different CLIP visual backbones. While larger models like ViT-L/14 yield the highest accuracy, the system achieves highly competitive results with lighter architectures (ViT-B/32, ViT-B/16), indicating that the method effectively extracts visual signals regardless of encoder capacity. All results are reported on the miniImageNet 5-way 1-shot task using a ResNet-12 backbone.

### A.4.5 EFFECT OF UNLABELED SAMPLES

In this section, we analyze how our results vary with the availability of unlabeled samples. As we increase the number of unlabeled samples, we observe a corresponding improvement in performance, which is intuitive (see Tables 12 and 13). We also compare our method with Cluster-FSL (Ling et al., 2022), which represents the previous state-of-the-art architecture in semi-supervised few-shot learning prior to our work.

### A.4.6 QUALITY OF REPRESENTATION SPACE AND PROTOTYPE REFINEMENT

Prior methods in SSFSL often commit to either global prototype-based structures (e.g., ProtoNet Snell et al. (2017), Soft K-Means Ren et al. (2018)) or reconstruction-based alignments (e.g., MFC Ling et al. (2022)). Prototype-based methods, while offering clear global decision boundaries, can falter in capturing fine-grained intra-class variations and local geometric nuances. Conversely, reconstruction-

based clustering excels at modelling local manifolds but often mis-assigns samples when two class subspaces overlap, because they ignore where class centroids sit in the broader topology. We overcome this dichotomy by fusing the reconstruction based distance with explicit Euclidean regularizers: an intra-class term that contracts each cluster and an inter-class term that repels different centroids. A point is accepted by a class only if it is simultaneously well reconstructed by that class's subspace and lies near its prototype while far from others. This fusion retains the fine-grained shape of each class manifold while simultaneously positioning the manifolds far enough apart to maintain clear global separation. The cluster separation tuner (CST) also adjusts each centroid to maximize intra-class compactness and inter-class separation. Applying this clustering method during the fine-tuning phase enhances the quality of the model's learned representations.

In the testing phase, we nudge each visual prototype a small step toward the CLIP (Radford et al., 2021) text embedding of its class description using our semantic injection network. This language-based anchor acts like a compass: it steers prototypes away from visually misleading regions (e.g., background clutter or occlusion) and pulls them toward positions that reflect the class's true meaning. That is particularly valuable when we have only a handful of labeled examples. During inference, CVOC coupled with semantic anchoring enables the identification of superior class prototypes which is conceptually aligned with the class description. This enhanced discriminability yields substantially higher pseudo-label accuracy than prior approaches (see Table 5, where we compared our method against Kmeans and MFC (Ling et al., 2022)), which in turn drives significant gains in FSL performance.

## A.5 Ablation Experiments

### A.5.1 Rationale for Pretraining and Finetuning

In our approach, pretraining learns a robust, general feature extractor from base classes, while finetuning adapts the model to few-shot episodes incorporating our novel components like CVOC. This clear separation ensures effective representation learning and task-specific adaptation. Empirically, skipping pretraining drops accuracy drastically (e.g., from 84.5% to 27% on MiniImageNet 5-way 1-shot with ResNet-12), and omitting finetuning also reduces performance significantly (to about 74.7%). These results highlight the essential roles of both stages in achieving strong few-shot performance.

### A.5.2 Results on Inductive Setup

Although our primary experiments were conducted under the transductive setting, we also adapt the algorithm to evaluate it in the inductive setup during testing. The results presented in Table 14 demonstrate that our method maintains strong performance, highlighting its robustness in this scenario as well.

In the *inductive setting*, the model must classify each query sample independently using only the labeled support set, without accessing other unlabeled query samples during prediction. In contrast, the *transductive setting* allows the model to jointly leverage the entire unlabeled query set at test time, often enabling label propagation or graph-based refinement.

In short: inductive FSL predicts each query in isolation, while transductive FSL predicts them jointly using information from the whole query distribution.

### A.5.3 Ablation and Sensitivity Analysis of CVOC and CST Hyperparameters

In this section, we ablate various hyperparameters used in the CVOC algorithm coupled with CST iterations in the finetuning stage. In the Figure 11, we present a heatmap analysis of the hyperparameters $w_{\text{intra}}$ and $w_{\text{inter}}$; while Table 4 reports results with and without incorporating intra and inter class terms. We defined the grid search over the following hyperparameter ranges:

- $\epsilon \in \{1.0, 2.0, 3.0\}$
- $\beta_0 \in \{0.01, 0.05, 0.07\}$
- $\gamma \in \{0.001, 0.003, 0.005, 0.01\}$
- $\alpha \in \{0.01, 0.02, 0.03\}$

Table 14: Ablation results under the inductive evaluation setup on the miniImageNet dataset using a ResNet-12 backbone.

| Setup | Acc (Ours) | Cluster-FSL (Ling et al., 2022) |
|---|---|---|
| *5-way-1-shot* | | |
| Transductive | 84.51 | 77.81 |
| Inductive | 85.68 | 78.83 |
| *5-way-5-shot* | | |
| Transductive | 86.95 | 85.55 |
| Inductive | 87.23 | 85.64 |

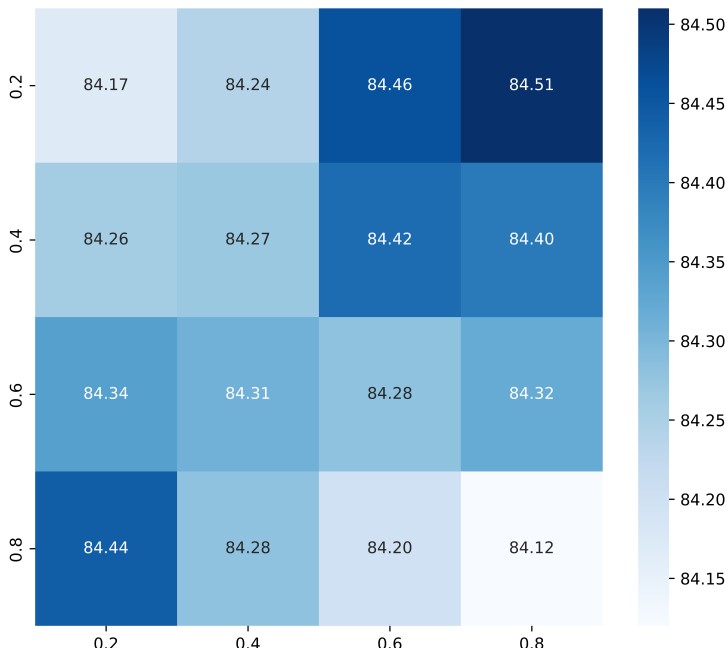

Figure 11: Heatmap showing the effect of $w_{\text{intra}}$ and $w_{\text{inter}}$ hyperparameters in the 5-way-1-shot setting and ResNet-12 backbone on *mini*ImageNet dataset. Here, the values of $w_{\text{intra}}$ vary row-wise, while the values of $w_{\text{inter}}$ vary column-wise.

The standard deviation of the downstream accuracy across the grid search was $\sigma = 0.097$, indicating stable results. Refer to Figure 4 to see results with and without CST updates.

### A.5.4 ABLATION ON THE CONTRIBUTIONS OF THE CLUSTER SEPARATION TUNER AND SEMANTIC CORRECTION

In this section, in order to isolate the contributions of the Cluster Separation Tuner (CST) and semantic correction, we disable the reconstruction-distance term and plug CST + semantic anchor directly into standard clustering methods. Here, we compare against the latest methods under the 5-way 1-shot and 5-shot tasks. Although we had to retune a few hyper-parameters - chiefly $s$ in the semantic anchor module and the number of CST iterations to fit this plug-in setup, the adjusted settings still yield competitive performance (Figure 8). We further incorporate Cluster-FSL equipped with Semantic Injection to clearly isolate its individual effect and to emphasize the modularity of the semantic injection mechanism.

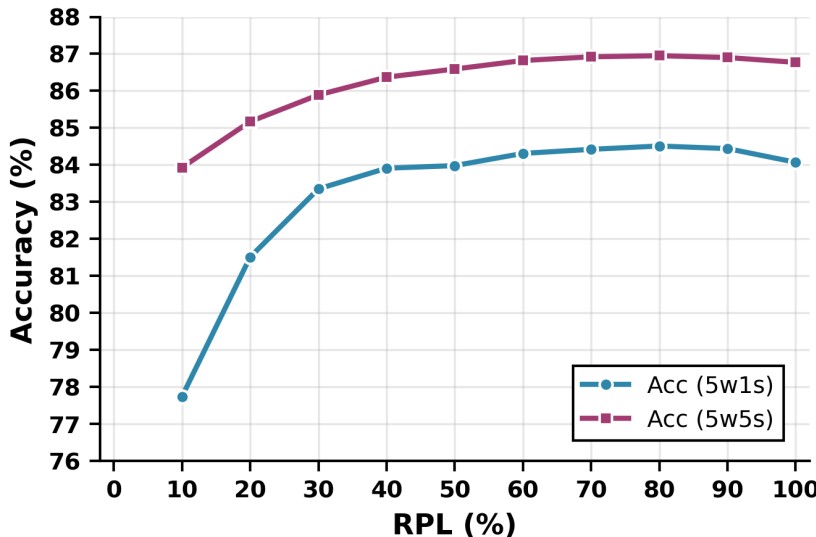

Figure 12: Accuracy on miniImageNet as a function of RPL (%) - the proportion of unlabeled samples retained based on lowest-entropy pseudo-labels. The *5w1s* and *5w5s* curves correspond to 5-way 1-shot and 5-way 5-shot settings, respectively, using a ResNet-12 backbone.

### A.5.5   ABLATION ON HYPERPARAMETER IN SEMANTIC ANCHOR

In the semantic anchor module, we introduce an extra hyperparameter ($s \in [0, 1]$; as discussed in the main paper) that controls the strength of shifting visual prototypes toward semantically plausible regions, helping to alleviate issues such as occlusions. Experimentally, this module proves especially beneficial in the 5-way 1-shot setting, likely because a single support example is more susceptible to errors when that image is occluded or otherwise ambiguous. As shown in Figure 9, accuracy steadily increases with $s$, peaks at $s = 0.9$, and then declines at $s = 1$.

### A.5.6   ABLATION ON RESTRICTED PSEUDO LABELING

At test time, we limit the augmentation of the support set by selecting only those unlabeled samples with the most confident pseudo-labels. Since pseudo-labels can be noisy, we sort unlabeled samples by their prediction entropy and pick the lowest-entropy examples, offering a simple, efficient strategy with minimal test-time overhead (see Figure 12). We observe that retaining 80% of the unlabeled samples yields the highest gains, highlighting the impact of confident pseudo-label selection. RPL performs well in every clean settings but may be less effective with distractor classes and very small unlabeled pools (<30) due to higher pseudo-label variance. However, with more than 50 unlabeled samples per class, it consistently improves accuracy even in distractor scenarios, while maintaining low inference overhead.

### A.5.7   CHOICE OF EMBEDDING MODEL

We evaluate two text encoders for generating class-description embeddings: a vanilla BERT-base model (Devlin et al., 2019) and the CLIP text encoder (Radford et al., 2021). In both 5-way 1-shot and 5-way-5-shot settings, CLIP's text features outperform BERT significantly (see Figure 13).

Potential reasons might be that CLIP's text encoder is trained jointly with its image encoder on millions of image–caption pairs via a contrastive objective, grounding its representations in visual concepts. BERT, by contrast, learns only linguistic co-occurrence patterns and lacks alignment with vision. Consequently, CLIP's text vectors naturally become a far more effective option for bridging semantic gaps in few-shot vision tasks than a standalone language model.

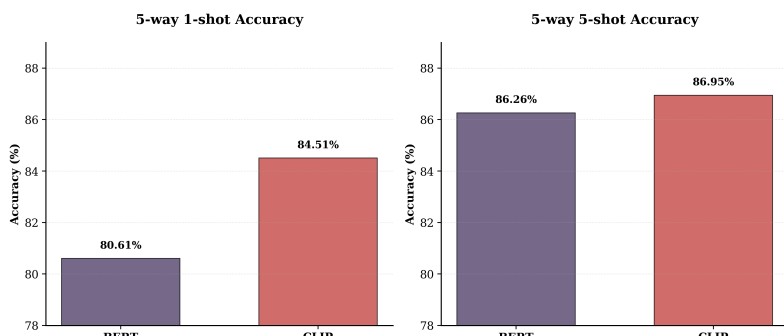

Figure 13: Accuracy comparison between BERT-base and CLIP text encoders on miniImageNet in 5-way 1-shot and 5-way 5-shot settings using ResNet-12 backbone. CLIP's multimodal pretraining yields consistently higher few-shot accuracy.

### A.5.8 ROBUSTNESS AND GENERALIZATION OF THE CORE VISUAL FRAMEWORK

Even though our class description generation pipeline and semantic anchor network are major contributions, we conducted a series of rigorous experiments analyzing the performance of our method in the absence of any semantic guidance. These analyses demonstrate that the visual-only components alone establish a powerful and generalizable foundation for semi-supervised few-shot learning.

**Ablation of Semantic Anchoring.** First, we evaluated our approach without the semantic anchoring module. As shown in the second row of our ablation study (Table 4), our method still outperforms recent state-of-the-art baselines.

**Cross-Domain Generalization.** A critical test for any few-shot learning method is its ability to generalize to entirely unseen domains. We assessed the generalization capability of our core visual framework by training and fine-tuning it on *mini*ImageNet (Vinyals et al., 2016) and subsequently testing it on four diverse target datasets: CUB (Hilliard et al., 2018), EuroSAT (Helber et al., 2019), ISIC (Codella et al., 2019), and ChestX (Wang et al., 2017), all without semantic anchoring. As detailed in A.1.2, the framework demonstrates remarkably strong performance, validating that its learned structural priors are not confined to the source domain but transfer effectively to new visual concepts.

**Qualitative Analysis of Class Activation.** Finally, we analyzed class activation maps (CAM) as described in A.1.3. We generated heatmaps for images from the CUB dataset using a model trained only on *mini*ImageNet. Compared to Cluster- FSL (Ling et al., 2022), our framework's heatmaps are more precisely localized on the object of interest, ignoring background clutter. This provides visual evidence that our methodology guide the model to focus on the true discriminative features of a class, leading to a more robust and interpretable representation space even in challenging cross-domain scenarios.

### A.6 ADDITIONAL RESULTS ON MODERN ARCHITECTURES (VIT & SWIN)

To further demonstrate the generalizability of our method beyond standard CNN backbones, we evaluated SA-CVOC on modern Vision Transformer architectures, specifically ViT-Small (Dosovitskiy et al., 2021) and Swin-Tiny (Liu et al., 2021). As shown in Table 15, our method consistently outperforms the strong baseline Cluster-FSL (Ling et al., 2022) across both architectures and settings on the *mini*ImageNet dataset. The work of Dosovitskiy et al. (2021) introduced the Vision Transformer, while Liu et al. (2021) proposed a hierarchical version. Our results show significant gains on Swin-Tiny, with an improvement of **+6.7%** in the 1-shot scenario and **+1.09%** in the 5-shot scenario, validating that our geometric regularization (CVOC) and semantic alignment modules are effective regardless of the underlying inductive bias (CNN vs. Transformer).

Table 15: Classification accuracy (%) on *mini*ImageNet using Vision Transformer backbones.

| Backbone | Method | 1-shot | 5-shot |
|----------|--------|--------|--------|
| ViT-Small | Cluster-FSL | 77.90 | 86.15 |
| | **Ours** | **84.20** | **87.21** |
| Swin-Tiny | Cluster-FSL | 79.33 | 88.31 |
| | **Ours** | **86.03** | **89.40** |

## A.7 LIMITATIONS

Our approach assumes that both labeled and unlabeled data lie on a shared manifold, which is crucial for accurate pseudo-labeling. Our semantic anchoring technique relies on the choice of embedding model used to encode class descriptions. Consequently, this semantic anchor module may not work very well in complex datasets where the visual features of different classes are highly similar or where the visual data lacks distinctive semantic characteristics necessary for effective differentiation.

