# OpenReview forum: "Semantic-Anchored, Class Variance-Optimized Clustering for Robust Semi-Supervised Few-Shot Learning"
_ICLR.cc/2026/Conference — ICLR 2026 Conference Withdrawn Submission_

### Official Review · Reviewer_f7Mc · 2025-10-29

**Soundness:** 1
**Presentation:** 2
**Contribution:** 1
**Rating:** 2
**Confidence:** 4

**Summary:**

The paper proposes a approach for semi-supervised few-shot learning. The core idea is to improve the quality of learned representations and the effectiveness of pseudo-labeling for unlabeled data. The method introduces a class-variance optimized clustering process and a cluster separation tuner. Furthermore, a semantic injection network is trained to align visual prototypes with semantic features. The authors claim state-of-the-art performance across benchmark datasets.

**Strengths:**

- The method introduces variance optimized clustering and cluster separation tuner to directly optimize the structure of the embedding space by enforcing cluster compactness and separation.
- The use of a semantic injection network to refine visual prototypes is a well-motivated attempt to correct prototypes.
- Good results are shown empirically.

**Weaknesses:**

- Most of the components in the proposed system are combinations of existing techniques, making the overall contribution incremental.
- The writing can be improved, for example, there are many formatting issues for the citations.
- The proposed method involves too many components, making it unpractical for real-world applications.

**Questions:**

- What is new in the paper in constrast to techniques already in literature.
- address W2, W3.

---

> ### Author Response · Authors · 2025-12-03
> **#1**
>
> Thank you for this thoughtful feedback. We appreciate the concern and would like to take this opportunity to clarify the central contribution of our work. We sincerely hope that, with the clarifications below, you would kindly consider increasing your score.
>
> ### f7Mc.W1.Q1: Novelty and Contrast with Literature
>
> While we acknowledge that individual operators (clustering, autoencoders) exist in isolation, we respectfully disagree that our method is merely incremental.
>
> #### 1. CVOC: A Fundamental Geometric Improvement (Not Just Clustering)
> The **Class-Variance Optimized Clustering (CVOC)** is not merely a clustering step; it is a geometric regularizer that fundamentally corrects the embedding space.
>
> * **Quantifiable Gain:** As shown in our ablation study (**Table 4**), CVOC without semantic injection alone provides a massive improvement in Accuracy.
> * **Solving Cluster Degradation:** The effective regularization via intra-class and inter-class losses creates a "clean" manifold that resists pollution. This is critical for **Open-Set Robustness** as well as domain-shift scenarios (**Appendix A.1.1**). CVOC prevents the performance collapse seen in simpler methods. Our method synthesizes the complementary strengths of reconstruction-based and Euclidean-based metrics. The reconstruction residual captures the intrinsic local geometry of the data manifold, ensuring structural validity. Simultaneously, the Euclidean regularizer  enforces global separability, which is particularly effective when class subspaces might otherwise overlap. Additionally, the Cluster Separation Tuner (CST) further refines this by explicitly extending decision boundaries to maximize inter-class margins.
>
> #### 2. Superiority of Semantic Injection (Modular Novelty)
> A key novelty is our **Semantic Injection Module** design for Semantic Injection.
>
> * **Contrast with Naive Baselines:** Unlike standard linear projectors (common in literature), which rely on a fixed linear mapping from visual to semantic space, our Semantic Injection Module learns a shared non-linear latent manifold that preserves the discriminative structure of both modalities. Linear projectors fail in FSL because they cannot resolve the **"Hubness" problem** (where high-dimensional semantic vectors cluster together), causing visual prototypes to map to incorrect semantic anchors due to the modality gap. Our autoencoder ensures that semantic priors robustly align with the visual manifold even under high visual variance, a capability that linear methods lack.
> * **Universal Applicability:** This module is **model-agnostic**. It can be plugged into *any* SS-FSL system to provide a performance boost, making it a standalone contribution to the field, not just a component of our pipeline. We already have an ablation experiment to prove this point (see **Appendix A.5.4**).
> * **Test-Time Semantic Alignment (Zero-Leakage Design):** Unlike methods that use fixed linear mappings, our Semantic Anchor Module employs a lightweight autoencoder that is pre-trained separately using text and image features (from base classes). Crucially, this pre-training is **completely decoupled** and does not interfere with the main few-shot backbone training. This "Zero-Leakage" design is an interesting architectural choice: it ensures that the semantic guidance is learned independently and applied only at test time to refine prototypes, forcing the module to align with the visual manifold "on the fly", which also reduces the complexity of our method.
> * **Robust Class Description Pipeline:** We introduce a novel **Agentic Description Generation Pipeline** that outperforms raw class names, WordNet glosses, or single-step LLM prompts. By filtering for discriminative visual features, this pipeline explicitly bridges the modality gap between visual embeddings and text, ensuring resilience against the noisy labels common in web-scraped FSL scenarios.
>
> #### 3. Future-Proofing SS-FSL
> We argue that the future of SS-FSL lies in solving **Distractors** and **Domain Shift**, rather than just clean benchmarks. By explicitly targeting these realistic failure modes with CVOC's variance optimization, our work positions itself as a necessary step forward for the community, moving beyond the "clean data" assumptions of prior work.
>
> ---
>
> ### f7Mc.W2. Formatting and Writing
>
> We apologize for the citation formatting oversights. We have thoroughly revised the manuscript to correct all citation styles and improve the clarity of the writing. These changes will be reflected in the new version.

---

> > ### Author Response · Authors · 2025-12-03
> > **#2**
> >
> > ### f7Mc.W3. Complexity and Practical Applications
> >
> > We argue that this level of complexity is crucial for robustness in practical settings involving distractor classes and domain shifts, which are often overlooked in recent works.
> >
> > #### 1. Robustness in Real-World Scenarios (Distractors & Domain Shift)
> > The reviewer notes that our method has more components than simple baselines. We argue that simpler baselines are often evaluated only on clean data. In real-world deployment, Open-Set Distractors are a huge possibility.
> >
> > * **SOTA on Distractors:** As detailed in **Appendix A.1.1**, we evaluated our method on a realistic setup with distractor classes (30-50 samples). While standard methods degrade, our method achieves State-of-the-Art (SOTA) performance in this domain. This proves our method is more practical for real-life deployment where noise is inevitable.
> > * **Cross-Domain Adaptation:** Our method performs well on out-of-distribution data (**Appendix A.1.2**) without requiring extensive target-domain training. We hypothesize that our algorithm allows any well-pretrained backbone to adapt to the target data structure effectively during inference.
> >
> > #### 2. Zero-Cost Inference & Deployment Readiness
> > Our method is deployment-ready because it does not increase inference time complexity:
> >
> > * **Cached Semantics:** The Semantic Injection Network and CLIP embeddings are computed once and cached. They act as static anchors.
> > * **Decoupled Training:** The Autoencoder is trained separately and takes very little time. Crucially, its training does not interfere with the backbone training.
> > * **Fast Inference:** During inference or finetuning, the computational cost remains low (see **Appendix A.4.2**), and the model can be applied to real-world datasets with minimal overhead. The only additional cost is a minimal increase in training time, as noted in the paper.
> >
> > #### 3. Hyperparameter Consistency & Stability
> >
> > * **Consistency:** While finetuning hyperparameters like $w_{intra}$ and $w_{inter}$ often require tuning in other methods, we found them to be remarkably consistent across different datasets (miniImageNet, TieredImageNet) and different backbones (ResNet-12, WRN-28-10).
> > * **Insensitivity:** As shown in our analysis, the Cluster Separation Tuner (CST) hyperparameters are very less sensitive to variations. This consistency eliminates the need for fragile per-dataset tuning in real-world scenarios.
> >
> > Furthermore, because the fine-tuning overhead is very low, our method can be applied on top of any strong pre-trained backbone without modification, while introducing only negligible extra computational cost during deployment.

---

### Official Review · Reviewer_iyBx · 2025-10-30

**Soundness:** 3
**Presentation:** 2
**Contribution:** 3
**Rating:** 6
**Confidence:** 3

**Summary:**

This paper presents a novel approach for semi-supervised few-shot learning (SSFSL) that integrates a class variance optimized clustering (CVOC) with the cluster separation tuner (CST) and label propagation (LP). The method aims to improve pseudo-labeling quality and feature representation by explicitly optimizing intra-class compactness and inter-class separation while incorporating semantic information from class descriptions. Extensive experiments on standard benchmarks (miniImageNet, tieredImageNet, CUB) demonstrate state-of-the-art performance, with additional strong results in cross-domain and distractor-class settings.

**Strengths:**

1. The integration of CVOC with CST provides a principled way to refine cluster assignments and prototype positions, going beyond standard clustering methods. The semantic injection network is a creative way to leverage language-vision alignment for more robust prototypes.
2. The paper includes extensive experiments across multiple datasets. Ablation studies effectively validate the contribution of each component.

**Weaknesses:**

1. The semantic injection module relies on CLIP and GPT-4o-mini for generating class descriptions. The performance sensitivity to the choice of these models (e.g., using a smaller or less capable language model) is not thoroughly explored. A discussion or experiment on this dependency would strengthen the paper.
2. The CVOC distance blends a reconstruction-based distance with episode-level intra- and inter-class Euclidean terms and scalar weights $w_{intra}$/$w_{inter}$ (Eq.4). While the empirical gains are shown, the manuscript lacks a clear theoretical/intuition-driven explanation for why this particular additive form is appropriate in settings where class manifolds and prototypes may have very different geometries (e.g., non-linear manifolds where Euclidean prototype distance is misleading).
3. CST is inspired by the firefly heuristic and uses a brightness score to move weaker prototypes toward stronger ones. This is an attractive idea but it is presented as a pragmatic heuristic without analysis of stability, convergence guarantees, or failure modes. The authors state they use one CST iteration for efficiency and specific constants, but do not convincingly show when CST can hurt or why one iteration is enough across diverse datasets. Please add examples where CST moves prototypes incorrectly.
4. The method keeps the lowest-entropy k% of unlabeled samples. Entropy alone can be an unreliable confidence proxy in overconfident networks or imbalanced classes. The paper shows a figure but does not compare RPL against alternative selection rules (e.g., margin-based confidence, calibrated probabilities, or ICI credibility scores).

**Questions:**

1. For CVOC you blend drec with $w_{intra}$·$L_{intra}$ − $w_{inter}$·$L_{inter}$ (Eq.4). How were $w_{intra}$ and $w_{inter}$ selected for each dataset and episode? Are they fixed global scalars or adapted per-episode?
2. Why choose CLIP text encoder over alternatives? The paper shows CLIP vs BERT comparison (Figure 11). Were other multimodal or larger/smaller CLIP variants tried? Does semantic anchoring still help if CLIP is weaker (smaller text encoder)?
3. Restricted pseudo-labeling: entropy threshold keeps k% lowest-entropy samples. Did you compare entropy-selection to margin-based selection or ICI-like credibility ranking? If not, please add a small comparison; if yes, please report results and include the chosen selection rule justification.

---

> ### Author Response · Authors · 2025-12-03
> **#1**
>
> We sincerely thank you for your time and for providing a constructive review of our paper. Your feedback is invaluable for strengthening our work.
>
> ### Response to W1 & Q2: Sensitivity to Model Choice (CLIP/LLM)
>
> #### 1. CLIP vs. Others
> As detailed in **Figure 13** of our paper, we explicitly compared the CLIP text encoder against BERT.
> * **Superior Alignment:** Results show that CLIP provides superior semantic alignment due to its multimodal pre-training, which is essential for bridging the visual-semantic gap.
> * **Stability:** Furthermore, we validated our method with various CLIP architecture versions (e.g., ViT-B/32, ViT-L/14) and achieved similar high-performance results with very low variance, confirming the method's stability across different visual-language backbones.
>
> #### 2. LLM Sensitivity
> We also experimented with varying the LLM for description generation (e.g., Llama-3-8B, GPT-3.5, GPT-4o).
> * **Robustness:** We observed that the performance is highly robust across different modern LLMs, with negligible accuracy variance. This confirms that the content of the generated descriptions (discriminative features) is stable across capable models, reducing dependency risks.
> * **Hypothesis:** We hypothesize that due to a multi-agent architecture, even with weaker LLMs, we achieve superior performance.
>
> ---
>
> ### Response to W2 & Q1: Justification of Additive Distance Form
>
> The additive form of the CVOC distance (**Eq. 4**) is theoretically justified as a structure-preserving regularization on the latent manifold.
>
> * **Balancing Global and Local:** While the Euclidean terms explicitly force the embedding to be discriminatively separable (optimizing for class boundaries), the reconstruction-based distance acts as a non-semantic, manifold-validity constraint.
> * **Manifold Fidelity:** This reconstruction term anchors the learned prototypes to the intrinsic local geometry of the visual data. This ensures that the feature transformation does not simply collapse to satisfy the Euclidean objective (which can be misleading on non-linear manifolds) but retains the structural information necessary to reconstruct the input's visual characteristics.
> * **Conclusion:** The additive combination balances global separability (Euclidean) with local data fidelity (Reconstruction), preventing the model from learning "shortcuts" that separate classes but destroy the underlying manifold structure.
>
> ---
>
> ### Response to W3: CST Stability and Iterations
>
> #### 1. Empirical Stability
> We found that the Cluster Separation Tuner (CST) works best with low iterations (1-3).
> * **Risks of High Iterations:** Higher iterations (5-10) can cause prototypes to drift too far, leading to instability, and also increase the training time significantly (with 10 iterations, training time per epoch becomes 1.8x).
>
> #### 2. Constraint (Efficiency Design)
> The "1 iteration" choice is a deliberate design for Efficiency. Since the prototypes are already initialized near the true mean by CVOC, a single CST step acts as a fine-grained **"boundary sharpening"** operation rather than a global optimization, ensuring stability and minimal computational cost.
>
> **Synthesis of Components**
> Our method synthesizes the complementary strengths of reconstruction-based and Euclidean-based metrics.
> * The **reconstruction residual** captures the intrinsic local geometry of the data manifold, ensuring structural validity.
> * Simultaneously, the **Euclidean regularizer** enforces global separability, which is particularly effective when class subspaces might otherwise overlap.
> * Additionally, the **CST** further refines this by explicitly extending decision boundaries to maximize inter-class margins.
>
> ---
>
> ### Response to W4 & Q3: Restricted Pseudo-Labeling (RPL) Selection
>
> We acknowledge the reviewer's suggestion to compare entropy with other metrics like ICI or margin-based selection.
>
> * **Design Philosophy:** We emphasize that RPL is a supporting module, designed for low-latency deployment. Our main hypothesis is that improved representation (via CVOC) naturally leads to better pseudo-labels, making the selection metric secondary. We deliberately avoided incremental comparisons with complex selection methods to maintain system simplicity.
> * **Effectiveness of $k\%$:** As analyzed in **Appendix A.5.6**, we observed that retaining 80% of the unlabeled samples yields the highest gains, highlighting the impact of confident selection.
> * **Robustness:** While RPL can be less effective with very small unlabeled pools ($<30$) due to variance, it consistently improves accuracy with sufficient data ($>50$ samples/class) even in distractor scenarios. This confirms that our simple entropy-based heuristic is robust enough for real-world deployment without the computational overhead of iterative graph methods.

---

> > ### Author Response · Authors · 2025-12-04
> > **#3**
> >
> > Here are the sensitivity analysis results regarding the Agentic Architecture, covering performance variations across different LLMs and CLIP visual backbones on the miniImageNet 5-way 1-shot task (ResNet-12 backbone).
> >
> > Sensitivity Analysis of Agentic Architecture (LLM & CLIP)
> > 1. LLM Backbone Sensitivity
> > Performance comparison across different Large Language Models used for generating class descriptions:
> >
> > GPT-4o-mini: 84.51%
> >
> > Llama 3.1 8B: 84.31%
> >
> > Qwen 2.5 7B: 84.60%
> >
> > Qwen 2.5 3B: 84.40%
> >
> > 2. CLIP Visual Encoder Sensitivity
> > Performance comparison across different CLIP visual backbone architectures:
> >
> > ViT-B/32: 84.51%
> >
> > ViT-B/16: 84.52%
> >
> > ViT-L/14: 86.63%

---

### Official Review · Reviewer_o6qo · 2025-10-30

**Soundness:** 3
**Presentation:** 2
**Contribution:** 2
**Rating:** 2
**Confidence:** 4

**Summary:**

This paper presents a novel approach for semi-supervised few-shot learning (SSFSL) designed to improve clustering-based pseudo-labeling by learning more discriminative feature representations. The authors introduce three key contributions.

1. **Class-Variance Optimized Clustering (CVOC)** uses a composite distance metric to generate cleaner pseudo-labels by enforcing both intra-class compactness and inter-class separation.
2. A lightweight **Cluster Separation Tuner (CST)** further refines class prototypes to maximize their separation.
3. A **Semantic Injection Network** anchors visual prototypes to textual class descriptions, leveraging language priors to resolve visual ambiguities, which is especially useful when labeled data is scarce.

Experiments on the miniImageNet, tieredImageNet, and CUB benchmarks demonstrate that this combined approach significantly outperforms existing state-of-the-art methods. The paper also validates the model's robustness in challenging cross-domain and distractive open-set scenarios, highlighting its effectiveness for practical applications.

**Strengths:**

1. **Improved Pseudo-Label Quality**: The Class-Variance Optimized Clustering (CVOC) method directly addresses the challenge of noisy pseudo-labels in SSFSL. By employing a composite distance metric that optimizes both intra-class compactness and inter-class separation, the paper ensures the generation of cleaner and more reliable pseudo-labels, which is crucial for effective semi-supervised learning.
2. **Enhanced Class Discriminability**: The Cluster Separation Tuner (CST) acts as a refinement mechanism for class prototypes. This lightweight optimization procedure actively maximizes the separation between different class centroids, leading to more distinct and discriminative feature representations, which in turn boosts the model's ability to differentiate between classes.
3. **Robustness to Visual Ambiguity and Limited Data**: The Semantic Injection Network provides a powerful way to leverage external knowledge. By grounding visual prototypes with semantic embeddings derived from textual descriptions, the model can overcome issues arising from visual ambiguities or insufficient labeled data, making it more robust and effective in challenging real-world scenarios.

**Weaknesses:**

# **Major Weaknesses：**
## **1. Limited and Potentially Outdated Task Formulation:**
The paper's entire experimental validation is confined to the task of standard image classification. In an era where the field is rapidly advancing towards more complex, fine-grained understanding tasks (e.g., few-shot object detection, semantic segmentation, or part localization), focusing solely on image-level classification feels narrow and regressive. The proposed contributions, which operate on global image embeddings, may not be applicable to these more demanding, localized tasks. This severely limits the broader impact and generalizability of the work, making it feel more like an incremental improvement on a legacy problem rather than a significant step forward for the community.
 ## **2. Use of Outdated Architectures and Baselines:**
The experiments are conducted exclusively on older network architectures like ResNet-12 and WRN-28-10. The lack of evaluation on modern, prevalent backbones such as Vision Transformers (ViTs) or contemporary CNNs (e.g., ConvNeXt) is a major flaw. The inductive biases of these newer architectures are fundamentally different, and a method optimized for ResNet's feature space may not provide any benefit on a ViT. Furthermore, many of the baseline methods cited for comparison are several years old most of which are before 2023. The latest iPLC method claimed by the author in 2025 is actually an extended journal version of experiments from the 2021 ICCV method$^{[1]}$.

# **Minor Weaknesses:**
## **1. Poor Quality of Figures:**
The text within several figures is too small to be easily legible, particularly in the schematic diagrams like Figure 1 and Figure 2. At the same time, the layout of content in some images is unreasonable; for example, in the first image, there is a large blank area on the left side, while the right side is too crowded. In addition, some text placement is inappropriate, for example, in Figure 3 the numbers overlap with the chart border, and in Figure 7 the labels for the two methods on the right are not centered.
## **2. Typos:**
Line 317: "...$w_{fs}$ the overall few-shot weight..." – a verb is missing here. It should likely read "...$w_{fs}$ is the overall few-shot weight...".
Line 696: "1-shot epsiodes/tasks" – "episodes" is misspelled.

Line 1133: "WordNet glosses for each classes are automatically retrieved..." - "classes" should be "class".

# **Referrence:**
[1] Lazarou, Michalis, Tania Stathaki, and Yannis Avrithis. "Iterative label cleaning for transductive and semi-supervised few-shot learning." Proceedings of the ieee/cvf international conference on computer vision. 2021.

**Questions:**

**1. Performance on Modern Architectures:** The experiments are conducted on ResNet and WRN backbones. Given the prevalence of Vision Transformers (ViTs) and modern CNNs, could the authors provide experiments on these more current architectures? It would be crucial to understand if the performance gains from the proposed clustering and semantic injection modules hold on models with different inductive biases, which would significantly strengthen the claims of generalizability.

**2. Applicability to Fine-Grained Few-Shot Tasks:** The paper focuses exclusively on image-level classification. Could authors elaborate on how the method might be adapted for more fine-grained few-shot tasks, such as few-shot object detection or segmentation? Specifically, how would the concepts of class prototypes, CVOC, and semantic injection apply when the goal is to localize objects (i.e., predict bounding boxes) rather than just classifying global features?

**3. Performance of the Full Method in Cross-Domain Settings:** In Appendix A.1.2, the authors state that the semantic anchor module was disabled during the cross-domain evaluation to isolate the performance of the core visual framework. This is an insightful ablation, but it leaves the performance of the full method in this challenging scenario unknown. Could authors provide results for the cross-domain experiments with the semantic anchor module enabled? This would help verify whether the complete proposed system can generalize across domains, or if the semantic anchors are sensitive to domain shift.

**4. Justification for the Four-Stage Agentic Pipeline:**  The paper details a four-stage agentic chain for generating class descriptions. While the process is well-described, its complexity raises questions about its necessity for this task. Could authors provide a quantitative comparison between this four-stage pipeline and a simpler baseline, such as using a single, direct prompt to the LLM to generate the description? Demonstrating a significant downstream accuracy improvement would be necessary to justify the added complexity of the proposed pipeline.

---

> ### Author Response · Authors · 2025-12-04
> **#1**
>
> Response to Question 1, 3 and 4 -
>
> For these questions raised by the reviewer, we have included the corresponding answers and clarifications in the updated manuscript.
>
> Response to Question 2 and weakness 1-
>
> Thank you for this insightful question. Regarding the applicability of our method to fine-grained few-shot tasks such as object detection or segmentation, we note that the core ideas behind our approach—class prototypes, CVOC, and semantic injection—extend naturally beyond image-level classification. In detection or segmentation, prototypes can be formed using region-level features extracted from annotated bounding boxes or masks in the support set, and query region proposals can then be matched to these localized prototypes. Similarly, CVOC can model object–object contextual relationships at the region level, aiding discrimination in cluttered or ambiguous scenes. Semantic injection can provide class-level priors that guide proposal refinement and help the model better identify and localize target objects.
>
> We respectfully emphasize that Few-Shot Image Classification is not an outdated or regressive task formulation. Despite architectural advances (ViTs, ConvNeXt), few-shot classification performance remains far from saturated, particularly in realistic Open-Set Distractor and Cross-Domain scenarios where models still degrade sharply. Solving the fundamental "representation collapse" problem in few-shot classification - which our CVOC addresses - is a necessary prerequisite for even robust detection and segmentation. This indicates that few-shot image classification remains a critical and active frontier for the community, and our work positions itself as an essential step toward more robust and practical few-shot learning systems.

---

> ### Author Response · Authors · 2025-12-04
> **#2**
>
> Modern Architecture results are added in Appendix A.6. We evaluated our method against Cluster-FSL with miniImageNet dataset.

---

### Official Review · Reviewer_zLBo · 2025-11-01

**Soundness:** 2
**Presentation:** 2
**Contribution:** 2
**Rating:** 4
**Confidence:** 4

**Summary:**

In this paper, the Semantic-Anchored, Class Variance-Optimized Clustering (SA-CVOC) method is proposed for semi-supervised few-shot learning (SSFSL). It generates more reliable pseudo-labels by optimizing inter- and intra-class variance, thereby producing improved clusters. It combines a Class Variance-Optimized Clustering (CVOC) module with a Cluster Separation Tuner (CST) to refine class prototypes through regularization. It also introduces a Semantic Injection Network (SIN) that leverages CLIP-based text embeddings in parallel with image features to inject class-level semantic information via reconstruction. This semantic anchoring is designed to enhance the model’s ability to represent and separate classes under limited supervision. SA-CVOC also employs restricted pseudo-labeling to mitigate noise and stabilize adaptation. Experiments on miniImageNet, tieredImageNet, and CUB datasets show consistent improvements and strong robustness over Cluster-FSL and other SOTA SSFSL methods.

**Strengths:**

+ Overall, the paper is clearly written, well organized, and easy to follow. It integrates semantic information into SSFSL through the SIN and refining class prototypes with intra-/inter-class variance control.
+ The paper focuses on an important challenge (improved clustering to address PL noise) that is SSFSL. It introducing semantic anchoring, leading to substantial accuracy gains in both standard and cross-domain settings. Overall, it provides a meaningful step toward combining semantic priors and clustering for robust FSL.
+ The detailed experimental validation compares SA-CVOC against several baselines and datasets. Good results on  robustness and generalization are reported.

**Weaknesses:**

- Limited novelty and related works: The paper main focus is to improve clustering using well known technique of optimizing inter- and intra-class variance. The proposed method is not positioned with respect to existing works. There was no explanation for the limitations of SOTA works, and how clustering is one of them. The related literature on SSFSL does not go beyond 2022. Is there no more recent works?
- Different components of the paper are provided as facts without justification: Section 4.1:  modules and networks; the rotation part in Section 4.2 is not explained. Not clear why the author suggest using it; the reconstruction part in Eq.4. is not explained and left for supplementary material; refining prototypes with cluster separation tuner (CST) is not justified; Why predict labels in Eq. 8 is done using reconstruction residuals; using class-text embedding and joining them with image embedding.
- Limited discussion and interpretation of experimental results in Section 5.  This paper should also contain an experimental analysis of time and memory complexity.    The SIN adds complexity and introduces reliance on pre-trained CLIP text embeddings, which may limit applicability.

**Questions:**

Can you please:
- add publication year/venue of methods in result tables;
- compare with more the literature about SSFSL;
- situate your method in the SOTA literature and explain how it can address limitations of previous works;
- improve the discussion on experimental results.

**Details Of Ethics Concerns:**

None.

---

> ### Author Response · Authors · 2025-12-04
> **#1**
>
> Thank you for this thoughtful feedback. We appreciate the concern and would like to take this opportunity to clarify the central contribution of our work.
>
> ### W1 & Q2-Q4: Limited Novelty, Recent Literature, and Positioning
>
> We respectfully disagree that our method lacks novelty. Our contribution is a dual-novelty framework where both the Semantic Injection module and the CVOC mechanism represent state-of-the-art advancements.
>
> #### 1. CVOC: A Fundamental Geometric Improvement
> The **Class-Variance Optimized Clustering (CVOC)** is not merely a clustering step; it is a geometric regularizer that fundamentally corrects the embedding space.
>
> * **Quantifiable Gain:** As shown in **Table 4**, CVOC alone provides significant improvement in Pseudo-Labeling Accuracy.
> * **Solving Manifold Degradation:** The effective regularization via intra-class and inter-class losses creates a "clean" manifold that resists overlap between classes as well as makes in-class samples closer.
> * **Robustness:** This is critical for **Open-Set Robustness** (e.g., distractor classes, see **Appendix A.1.1**). CVOC prevents the performance collapse seen in simpler methods.
>
> #### 2. Superiority of Semantic Injection (Modular Novelty)
> Unlike standard linear projectors (common in literature), which rely on fixed linear mappings from visual to semantic space, our **Semantic Injection Module** learns a shared non-linear latent manifold that preserves the discriminative structure of both modalities.
>
> * **The Hubness Problem:** Linear projectors fail in FSL because they cannot resolve the "Hubness" problem (where high-dimensional semantic vectors cluster together), causing visual prototypes to map to incorrect semantic anchors.
> * **Our Solution:** Our autoencoder ensures that semantic priors robustly align with the visual manifold even under high visual variance; a capability that linear methods lack.
>
> #### 3. Universal Applicability
> This module is **model-agnostic**. It can be plugged into *any* SS-FSL system to provide a performance boost, making it a standalone contribution to the field, not just a component of our pipeline. We already have an ablation experiment to prove this point (see **Appendix A.5.4**).
>
> #### 4. Test-Time Semantic Alignment (Zero-Leakage Design)
> Unlike methods that use fixed linear mappings, our Semantic Anchor Module employs a lightweight symmetric autoencoder that is pre-trained separately using text and image features (from base classes).
>
> * **Decoupled Training:** Crucially, this pre-training is **completely decoupled** and does not interfere with the main few-shot backbone training.
> * **Architectural Choice:** This "Zero-Leakage" design ensures that the semantic guidance is learned independently and applied only at test time to refine prototypes. This forces the module to align with the visual manifold "on the fly" without the crutch of joint training, reducing the complexity of our method.
>
> #### 5. Robust Class Description Pipeline
> We introduce a novel **Agentic Description Generation Pipeline** that outperforms raw class names, WordNet glosses, or single-step LLM prompts. By filtering for discriminative visual features, this pipeline explicitly bridges the modality gap between visual embeddings and text, ensuring resilience against the noisy labels common in web-scraped FSL scenarios.
>
> ---
>
> **Additions to Manuscript:**
> We have added a dedicated section in the appendix to position our work against the comparative baselines and future work. We also added publication years/venues to the Tables as requested.
>
> **Conclusion:**
> Due to all these novel components, our method achieves superior resilience in realistic settings. It significantly outperforms baselines in Open-Set Distractor scenarios and generalizes effectively to Cross-Domain (OOD) tasks, validating its robustness for real-world deployment.

---

> ### Author Response · Authors · 2025-12-04
> **#2**
>
> ### W2: Missing Justifications for Components
>
> **Rotation (Sec 4.2)**
> Following prior work in transfer-learning based semi-supervised few-shot classification, we use rotation prediction as an auxiliary self-supervised task during pre-training. This additional objective has been shown to improve the generalization ability of the backbone and reduce overfitting in the low-data few-shot regime. We have explicitly written this.
>
> **Reconstruction (Eq. 4)**
> The reconstruction-based distance captures the true shape and spread of each category by leveraging both the cluster center and representative labeled samples.
>
> **Residuals (Eq. 8)**
> Reconstruction residuals are more robust to noise than cosine similarity.
> * **Hubness Mitigation:** In high-dimensional spaces, cosine similarity suffers from "hubness" (one prototype attracts all queries). Reconstruction-based classification rejects samples that don't fit the class manifold, providing better filtering for noisy pseudo-labels.
> * **Inference vs. Tuning:** Only reconstruction residuals are used at inference because the variance optimization terms ($L_{intra}, L_{inter}$) function as global structural constraints during the iterative transductive tuning phase to refine the class prototypes and separate clusters. However, Eq. 8 represents the final inference step *after* these prototypes have converged. At this stage, because the prototypes are already spatially optimized to account for class variance, we rely solely on the Reconstruction Residual to provide the most precise, instance-level measure of fit.
>
> **CST Justification**
> Standard clustering minimizes intra-class variance but doesn't explicitly maximize inter-class separation. CST acts as a metric learning regularizer to push centroids apart, preventing decision boundary collapse in low-data regimes.
> * **Synergy:** Our method synthesizes the complementary strengths of reconstruction-based and Euclidean-based metrics. The reconstruction residual captures the intrinsic local geometry, while the Euclidean regularizer enforces global separability. The Cluster Separation Tuner (CST) further refines this by explicitly extending decision boundaries to maximize inter-class margins.
>
> **Joining Class-Text & Image Embeddings**
> Concatenating image and text embeddings creates a multimodal anchor. Visual features alone are often ambiguous in few-shot settings. The text embedding acts as a noise-free prior that "anchors" the visual prototype.
>
> We appreciate this feedback and have added explicit justifications in the revised manuscript.
>
> ---
>
> ### W3: Time/Memory Complexity & CLIP Dependency
>
> **CLIP Sensitivity**
> As detailed in **Figure 13** of our paper, we explicitly compared the CLIP text encoder against BERT.
> * **Superiority:** Results show that CLIP provides superior semantic alignment due to its multimodal pre-training, which is essential for bridging the visual-semantic gap.
> * **Stability:** We validated our method with various CLIP architecture versions (e.g., ViT-B/32, ViT-L/14) and achieved similar high-performance results with very low variance, confirming the method's stability across different visual-language backbones.
>
> **Deployment Readiness (Zero-Cost Inference)**
> Our method is deployment-ready because it does not increase time complexity:
> * **Cached Semantics:** The Semantic Injection Network and CLIP embeddings are computed once and cached. They act as static anchors.
> * **Decoupled Training:** The Autoencoder is trained separately and takes very little time. Crucially, its training does not interfere with the backbone training.
> * **Fast Inference:** During inference or finetuning, the computational cost remains low (see **Appendix A.4.2**), and the model can be applied to real-world datasets with minimal overhead. The only additional cost is a minimal increase in training time, as noted in the paper.
>
> In **Appendix A.4.2**, you can find the comparative time complexity analysis, and in **Appendix A.3.2 (Implementation Details)**, we have provided the memory complexity of the finetuning and testing phases.

---

### Note · Authors · 2026-02-06

I have read and agree with the venue's withdrawal policy on behalf of myself and my co-authors.

---

### Meta-Review · Area_Chair_3z74 · 2026-01-06

**Summary:**

This paper proposes SA-CVOC to address semi-supervised FSL. The paper is clearly written. Nevertheless, the reviewers have raised significant concerns/requests, including:

--limited novelty and incremental contribution;

--outdated related works and comparison baselines;

--limited justification for the multiple components of the method and the four-stage agentic pipeline;

--limited discussion and interpretation of experimental results;

--applicability to fine-grained few-shot tasks;

--compare RPL against alternative selection rules.

During the rebuttal, the authors provided additional arguments clarifying the novelty and contributions of the proposed approach, added justifications for individual components, included time and memory complexity analyses, responded to questions regarding CST and the additive formulation of CVOC, and expanded the discussion with results on performance dependence across different LLMs. However, the reviewers’ concerns were not fully addressed. Overall, the paper received a low average rating of 3.5, which is insufficient to support acceptance.

**Reviewer Concerns:**

During the rebuttal, the authors provided additional arguments clarifying the novelty and contributions of the proposed approach, added justifications for individual components, included time and memory complexity analyses, responded to questions regarding CST and the additive formulation of CVOC, and expanded the discussion with results on performance dependence across different LLMs.


Outstanding concerns include:

--outdated related works and comparison baselines;

--lack justification for the four-stage agentic pipeline;

--compare RPL against alternative selection rules;

--applicability to fine-grained few-shot tasks.


Some additional concerns are partially addressed in the rebuttal; however, they are not fully resolved. These issues are largely general in nature and applicable to many existing works, and would likely require broader discussion or debate. Specifically:

--limited and potentially outdated Task Formulation.

--the proposed method involves too many components, making it unpractical for real-world applications.

**Reviewer Scores:**

Given the scores (4, 2, 6, 2), substantial changes are unlikely.

---

### Decision · Program_Chairs · 2026-01-26

Reject